

# Unique spicules may confound species differentiation: taxonomy and biogeography of *Melonanchora* Carter, 1874 and two new related genera (Myxillidae: Poecilosclerida) from the Okhotsk Sea

Andreu Santín[1], María-Jesús Uriz[2], Javier Cristobo[3,4], Joana R. Xavier[5,6] and Pilar Ríos[3,7]

[1] Institut de Ciències del Mar (ICM-CSIC), Barcelona, Catalonia, Spain
[2] Centre d'Estudis Avançats de Blanes (CEAB-CSIC), Blanes, Catalonia, Spain
[3] Instituto Español de Oceanografía. Centro Oceanográfico de Gijón., Gijón, Asturias, Spain
[4] Departamento de Ciencias de la Vida, EU-US Marine Biodiversity Group, Universidad de Alcalá, Alcalá de Henares, Madrid, Spain
[5] CIIMAR–Interdisciplinary Centre of Marine and Environmental Research, University of Porto, Porto, Portugal
[6] Department of Biological Sciences, University of Bergen, Bergen, Norway
[7] Departamento de Biología Animal, Universidad de Málaga, Málaga, Spain

Corresponding author
Andreu Santín, santin@icm.csic.es

## ABSTRACT

Sponges are amongst the most difficult benthic taxa to properly identify, which has led to a prevalence of cryptic species in several sponge genera, especially in those with simple skeletons. This is particularly true for sponges living in remote or hardly accessible environments, such as the deep-sea, as the inaccessibility of their habitat and the lack of accurate descriptions usually leads to misclassifications. However, species can also remain hidden even when they belong to genera that have particularly characteristic features. In these cases, researchers inevitably pay attention to these peculiar features, sometimes disregarding small differences in the other "typical" spicules. The genus *Melonanchora* Carter, 1874, is among those well suited for a revision, as their representatives possess a unique type of spicule (spherancorae). After a thorough review of the material available for this genus from several institutions, four new species of *Melonanchora*, *M. tumultuosa* sp. nov., *M. insulsa* sp. nov., *M. intermedia* sp. nov. and *M. maeli* sp. nov. are formally described from different localities across the Atlanto-Mediterranean region. Additionally, all *Melonanchora* from the Okhotsk Sea and nearby areas are reassigned to other genera; *Melonanchora kobjakovae* is transferred to *Myxilla* (*Burtonanchora*) while two new genera, *Hanstoreia* gen. nov. and *Arhythmata* gen. nov. are created to accommodate *Melonanchora globogilva* and *Melonanchora tetradedritifera*, respectively. *Hanstoreia* gen. nov. is closest to *Melonanchora*, whereas *Arhythmata* gen. nov., is closer to *Stelodoryx*, which is most likely polyphyletic and in need of revision.

## INTRODUCTION

Accurate species-level taxonomy is a fundamental keystone for conservation assessment, planning, and management (*Myers et al., 2000*; *Groves et al., 2017*). The differentiation between cryptic species (as in *Knowlton, 1993*), is of paramount importance for effective conservation policies (*Lohman et al., 2010*). While cryptic species are a widespread phenomenon among both terrestrial and marine phyla (*e.g.*, *Baker, 1984*; *Mayer & Helversen, 2001*; *Concepción et al., 2008*; *Crespo & Pérez-Ortega, 2009*; *Dennis & Hellberg, 2010*; *Lohman et al., 2010*; *Payo et al., 2013*; *Golestani et al., 2019*), the assumed lack of barriers to gene flow in marine habitats (*Hellberg, 2009*) contributed to the assumption that benthic organisms have greater distribution ranges and phenotypic plasticity than their terrestrial counterparts (*Knowlton, 1993*). As a result of this assumption, many benthic species were considered to be geographically widespread or even cosmopolitan (*Klautau et al., 1999*). Recent studies have generally demoted this idea (e.g., *Klautau et al., 1999*; *van Soest, Hooper & Hiemstra, 1991*; *van Soest & Hooper, 1993*). The dispersal capabilities vary greatly among benthic species even within the same phyla (*Uriz et al., 1998*) and they can be differentially reduced by natural barriers (*Allcock et al., 1997*; *Waters & Roy, 2004*). In this sense, some invertebrate Phyla, such as sponges and corals, produce short-lived, free larvae that are seemingly incapable of countering apparently weak marine barriers such as littoral currents or substrate discontinuity, often resulting in extremely low dispersal capabilities (*Hellberg, 2009*). In sponges, for instance, genetically structured populations, even at short spatial scales, have been repeatedly reported (*Duran et al., 2004*; *Duran, Pascual & Turon, 2004*; *Calderon et al., 2007*; *Blanquer, Uriz & Caujapé-Castells, 2009*; *Blanquer & Uriz, 2010*; *Guardiola, Frotscher & Uriz, 2016*), which favours speciation and makes the existence of widely distributed or cosmopolitan species unlikely.

Species complexes and cryptic species are particularly prevalent among sponges with few diagnostic characters (*Klautau et al., 1999*; *Uriz, Garate & Agell, 2017a*, *2017b*), especially when these characters exhibit environmental plasticity (*Maldonado et al., 1999*; *Xavier et al., 2010*; *De Paula et al., 2012*). For example, the sponge complex *Chondrilla nucula Schmidt, 1862*, was once considered as having a circumtropical distribution (*Klautau et al., 1999*), *Stylocordyla borealis* (*Lovén, 1868*) was reported as occurring at both poles (*Uriz et al., 2010*), the Atlanto-Mediterranean *Scopalina lophyropoda Schmidt, 1862* and *Hemimycale columella* (*Bowerbank, 1874*) both contained several morphologically cryptic species revealed by molecular markers (*Blanquer & Uriz, 2008*; *Uriz, Garate & Agell, 2017a*, *2017b*) and the excavating sponges *Cliona celata Grant, 1826* and *Cliona viridis* (*Schmidt, 1862*), which are known to be "species complexes" which remain partially unresolved (*Xavier et al., 2010*; *De Paula et al., 2012*; *Escobar, Zea & Sánchez, 2012*; *Leal et al., 2016*; *Gastaldi et al., 2018*). Cryptic species complexes are also prevalent in sponge genera without mineral (spicules) or organic skeletons (spongin fibres), such as

*Hexadella Topsent, 1896*, where species are almost indistinguishable based solely on morphological or histological characteristics (*Reveillaud et al., 2010, 2012*). However, species can also remain hidden even when they belong to genera that have particularly characteristic spicules. In these cases, researchers inevitably pay attention to these peculiar spicules, sometimes disregarding small differences in the other "typical" spicules.

Some genera of Poecilosclerida, one of the most diverse orders in terms of spicule diversity (*Hooper & Van Soest, 2002*), possess unique spicular types that greatly facilitate their identification. Examples include dianciastras in *Hamacantha Gray, 1867* (*Hajdu, 1994*; *Hajdu & Castello-Branco, 2014*), clavidiscs in *Merlia* Kirkpatrick, 1908 (*Vacelet & Uriz, 1991*), discorhabds in *Latrunculia du Bocage, 1869* (*Samaai, Gibbons & Kelly, 2006*) or thraustoxeas in *Rhabderemia Topsent, 1890* (*van Soest & Hooper, 1993*). Nevertheless, because taxonomists historically have focused on these particular spicules (*van Soest, Hooper & Hiemstra, 1991*), differences in other apparently banal spicules have been disregarded. As a consequence, some of these genera (*e.g., Rhabderemia van Soest & Hooper (1993)*, *Acarnus, Gray, 1867*, *van Soest, Hooper & Hiemstra (1991)*, *Merlia, Vacelet & Uriz (1991)* or *Trachytedania Ridley, 1881* (*Cristobo & Urgorri (2001)*)) contain or contained, until recently, few formally described species that were considered as having a widespread geographic distribution. Moreover, only the well-described species are usually recognised and reported in the literature (*van Soest, Hooper & Hiemstra, 1991*), while those with poor or imprecise descriptions remain forgotten, a trend which is aggravated for sponges living in remote or hardly accessible environments, such as the deep-sea (*Reveillaud et al., 2010*). Despite the challenges involved, comprehensive reviews of such genera are considered extremely useful for the discovery of cryptic species (*Reveillaud et al., 2012*) and to test biogeographical and evolutionary hypotheses (*van Soest & Hooper, 1993*; *Cárdenas et al., 2007*).

The genus *Melonanchora Carter, 1874*, is among those well suited for such revisions, as (i) it possesses a unique spicule type (spherancorae); (ii) currently contains only five formally accepted species (*van Soest et al., 2021*) (iii) only two out of the five species are commonly recorded over large geographical areas (*Baker et al., 2018*) and (iv) the three remaining species seem to be endemic to the Okhotsk Sea and nearby Pacific Islands (*Koltun, 1958, 1970*; *Lehnert, Stone & Heimler, 2006a*) and present clear differences with their Atlantic counterparts (*Lehnert, Stone & Heimler, 2006a*). Finally, *Melonanchora* representatives occur within Vulnerable Marine Ecosystems (VMEs) across the Atlanto-Mediterranean region, thus being in need of accurate identifications for the evaluation of the conservation status of the sponge grounds where they occur (*Best et al., 2010*; *ICES, 2012*).

In this context, this paper: (1) reviews the status of all the species currently allocated to *Melonanchora* with particular emphasis in the Pacific species, apparently endemic to the Okhotsk Sea, and their relationships with other Myxillidae; (2) provides a reliable guide for their identification; (3) describes new species of the genus; (4) and discusses the biogeographical implications of the circumpolar distribution of this genus.

## MATERIALS AND METHODS

### Museum material and sample treatment

The materials for this study consisted of samples from natural history museums and other scientific institutions and unregistered individuals from surveys across the North Atlantic (Life+ INDEMARES, NEREIDA and ABIDES) as well as specimens from authors' own collections. The institutions are abbreviated in the text as follow:

Canadian Museum of Nature, Canada (CMNI), using the prefix CMNI; Gothenburg Natural History Museum, Sweden (GNM), using the prefix GNM; Museo Civico di Storia Naturale di Genova, Italy (MSNG), using the prefix MSNG; Museum of Biology of Lund, Sweden (MZLU), using the prefix MZLU; Naturalis Biodiversity Center, The Netherlands (NBC, previously ZMA), using the prefix ZMA.POR. and ZMA.POR.P; National Museum of Natural History, Smithsonian Institution, Unites States (NMNH, previously USNM) using the prefix NMNH-USNM; Musée Zoologique de la Ville de Strasbourg, France (MZS) using the prefix MZS; Museu de Ciències Naturals (Zoologia) de Barcelona, Spain (MZB), using the prefix MZB; National History Museum, United Kingdom (NHMUK, previously BMNH), using the prefix NHMUK; Swedish Museum of Natural History, Sweden (NRM), using the prefix NRM; Yale Peabody Museum of Natural History, Unites States (YPM), using the prefix YPM IZ; Museum für Naturkunde, Germany (ZMB) using the prefix ZMB; Jean Vacelet's personal collections (JV) and Manuel Solórzano's personal collections (MS).

DNA was extracted from small pieces of tissue of four samples (MSNG Vis4.7, CMNI 2018-0107, GNM Por624, NMNH-USNM 1082996) using QIAGEN's DNeasy Blood and Tissue kit, following the instructions of the manufacturer. Amplification and sequencing of the mitochondrial cytochrome c oxidase subunit I (COI) were attempted but proved unsuccessful, with only two samples yielding an amplicon but resulting in sequencing of non-target DNA (bacteria). This was likely due to the low quantity and integrity of the DNA in the samples, as assessed by spectrophotometry using a DeNovix DS-11 FX.

All known species of *Melonanchora* were represented in the studied material, with the holotypes for all species but *Melonanchora tetradedritifera Koltun, 1970* being examined. Spicule preparations for both optical and scanning electron microscopy (SEM) were performed according to *Cristobo et al. (1993)* and *Uriz, Garate & Agell (2017a)*. Optical observations were performed using a Leica DM IRB inverted microscope from the Instituto de Ciencias del Mar (ICM-CSIC), whereas SEM observation were conducted using an ITACHI TM3000 TableTop Scanning Electron Microscope from the Center for Advanced Studies of Blanes (CEAB-CSIC), Spain, a JEOL–6100 SEM from the University of Oviedo (UO), Spain, and a HITACHI S-3500 N scanning electron microscope from the Institut de Ciències del Mar (ICM-CSIC), Spain. Spicule sizes are given as ranges with average values (in italics) ± Standard Deviation (*e.g.*, MIN.–*MEAN* ± *SD*–MAX.). Unless otherwise stated, spicule measurements were performed on 40 spicules per spicule type. The species classification adopted in the study follows that currently proposed by *Morrow & Cárdenas (2015)* and the World Porifera Database (*van Soest et al., 2021*). A key to *Melonanchora* can be found in Supplemental Material 1.

Finally, the electronic version of this article in Portable Document Format (PDF) will represent a published work according to the International Commission on Zoological Nomenclature (ICZN), and hence the new names contained in the electronic version are effectively published under that Code from the electronic edition alone. This published work and the nomenclatural acts it contains have been registered in ZooBank, the online registration system for the ICZN. The ZooBank LSIDs (Life Science Identifiers) can be resolved and the associated information viewed through any standard web browser by appending the LSID to the prefix http://zoobank.org/. The LSID for this publication is: [urn:lsid:zoobank.org:pub:F1A22CAA-DE1F-434D-9A6B-F00853C40FF5]. The online version of this work is archived and available from the following digital repositories: PeerJ, PubMed Central SCIE and CLOCKSS.

# RESULTS

## Systematic Description

Phylum PORIFERA *Grant, 1836*
Class DEMOSPONGIAE *Sollas, 1885*
Subclass Heteroscleromorpha *Cárdenas, Pérez & Boury-Esnault, 2012*
Order POECILOSCLERIDA *Topsent, 1928*
Family MYXILLIDAE *Dendy, 1922*
Genus *Melonanchora Carter, 1874*

Type species:
*Melonanchora elliptica Carter, 1874*: 212 (by monotypy).

Diagnosis:
From encrusting to massive-globular growth form, with paper-like, easily detachable thin ectosome, bearing fistular processes. Ectosomal skeleton composed of smooth strongyles to tylotes with somewhat asymmetrical ends, whereas the choanosome is mainly composed of smooth strongyles or styles. Microscleres include typically two categories of anchorate isochelae, rarely three, and spherancorae (amended from *van Soest, 2002*).

Remarks:
The genus *Melonanchora* was erected by *Carter (1874)* for *Melonanchora elliptica* on the account of this species singular anchorate-derived chelae (spherancorae), placing it tentatively with the "*Halichondria*" family concept built around *H.* (= *Myxilla*) *incrustans* (*Johnston, 1842*). The genus was later included in Desmacidonidae *Schmidt (1880)* until *Lundbeck (1910)*, and later *Topsent (1928)*, transferred it to Myxillidae. Simultaneously, Hentschel had it assigned it to Dendoricellidae[1] (*Hentschel, 1929*), but this assignation was not widly accepted (*Alander, 1935*) and was quickly disregarded.

The family Myxillidae has been redefined over the years (*Hajdu, van Soest & Hooper, 1994*; *Desqueyroux-Faúndez & van Soest, 1996*; *van Soest, 2002*) and the genus *Melonanchora* fits well within the current definition of Myxillidae established in the Systema Porifera (*Hooper & Van Soest, 2002*), which is restricted to "those genera which combine the possession of anchorate chelae with diactinal ectosomal tornotes

---

[1] While Hentschel assigned it to Dendoricellidae, he later wrongfully referred *Melonanchora* as part of Tedanidae within the text.

[oxeotes and tylotes] and choanosomal styles in a reticulate arrangement". Yet, after re-examination of all the available *Melonanchora* material, the current definition of the genus (*van Soest, 2002*) needs to be amended to better allocate the new species here described or re-described, including: presence of smooth strongyles (*Melonanchora emphysema* (*Schmidt, 1875*), *M. tumultuosa* sp. nov., *Melonanchora intermedia* sp. nov.) as choanosomal megascleres and the possession of two to three chelae categories (*M. intermedia* sp. nov., *M. maeli* sp. nov.).

Nevertheless, the main diagnostic character of the genus, the spherancorae, remains unaltered since Carter's original description (See "The Origin of Spherancorae"). Aside from spherancorae, Carter also added the presence of a papillated paper-thin like ectosome (Figs. 1A, 1C, and 1F) as an additional diagnostic character (*Carter, 1874*). Although this feature is shared with other deep-sea genera such as *Cornulum Carter, 1876* or *Coelosphera Thomson, 1873* (*Lehnert & Stone, 2015*; *Schejter, Cristobo & Ríos, 2019*), *Melonanchora* differs from the later in its white-translucent coloration, brittle and loose appearance and its characteristic wart-shaped papillae, which may make external identification feasible at the genus level (*Stone, Lehnert & Reiswig, 2011*).

<p align="center">*Melonanchora elliptica* Carter, 1874<br>(Figs. 1A, 2, 3)</p>

Synonymy:
*Melonanchora elliptica* Carter, 1874: 212, pI. XIII figs 6–12, pI. XV figs. 35a–35b; *Vosmaer, 1885*: 31, pI. I fig. 14, pI. V figs. 69–70 (*partim*); *Topsent, 1892*: 101–102; *Fristedt, 1887*: 454, pl. 25 fig. 5, 55 (*partim*); *Arnesen, 1903*: 15–16, pl. II fig. 4, pl. V fig. 4; *Topsent, 1904*: 144, pl. IV fig. 10; *Lundbeck, 1905*: 213–216, pl. VII figs. 4–6, pl. XX figs. 1a–1o; *Lundbeck, 1909*: 402–403; *Arndt, 1913*: 116; *Topsent, 1913*: 44; *Topsent, 1928*: 246; *Hentschel, 1929*: 966; *Burton, 1931*: 4; *Alander, 1935*: 5; *Arndt, 1935*: 71–73, Fig. 141; *Koltun, 1959*: 122–123, fig. 76; *Ríos & Cristobo, 2017*: 169; *Baker et al., 2018*: 20–25, figs. 5–7; *Dinn & Leys, 2018*: 63.
Not: *M. elliptica*; *Schmidt, 1880*: 85, pl. IX fig. 8.

Material examined.

Holotype: NHMUK 1882.7.28.54a, between the north coast of Scotland and the Faroe Islands; *HMS Porcupine* expedition (1869), ca. 800 m depth, 1869. (two slides); NHMUK-Norman Coll. N°50 10.1.1.1417, *HMS Porcupine* expedition (1869); NHMUK 1954.3.9.301 N°50; NHMUK - Norman Coll. -H. J. Carter Slide Coll. 1954.3.9.301; ZMB Por 3042, between the North coast of Scotland and the Faroe Islands, North Atlantic Ocean (59.85166, −6.03333).

Additional specimens examined:
CMNI 2018-0107, Saglek Bank, Labrador Sea, North Atlantic Ocean (60.45213, −61.26894), 427 m depth, 2016-07-21, collected by Dinn, Curtis (*Dinn & Leys, 2018*); MZLU L936/3483, Trondheim Fjord, Norway (63.494092, 10.31647), 1936; NRM 113070, off Lindenows Fjord, Greenland, North Atlantic Ocean (60.06666, −34.25), 237.9 m depth,

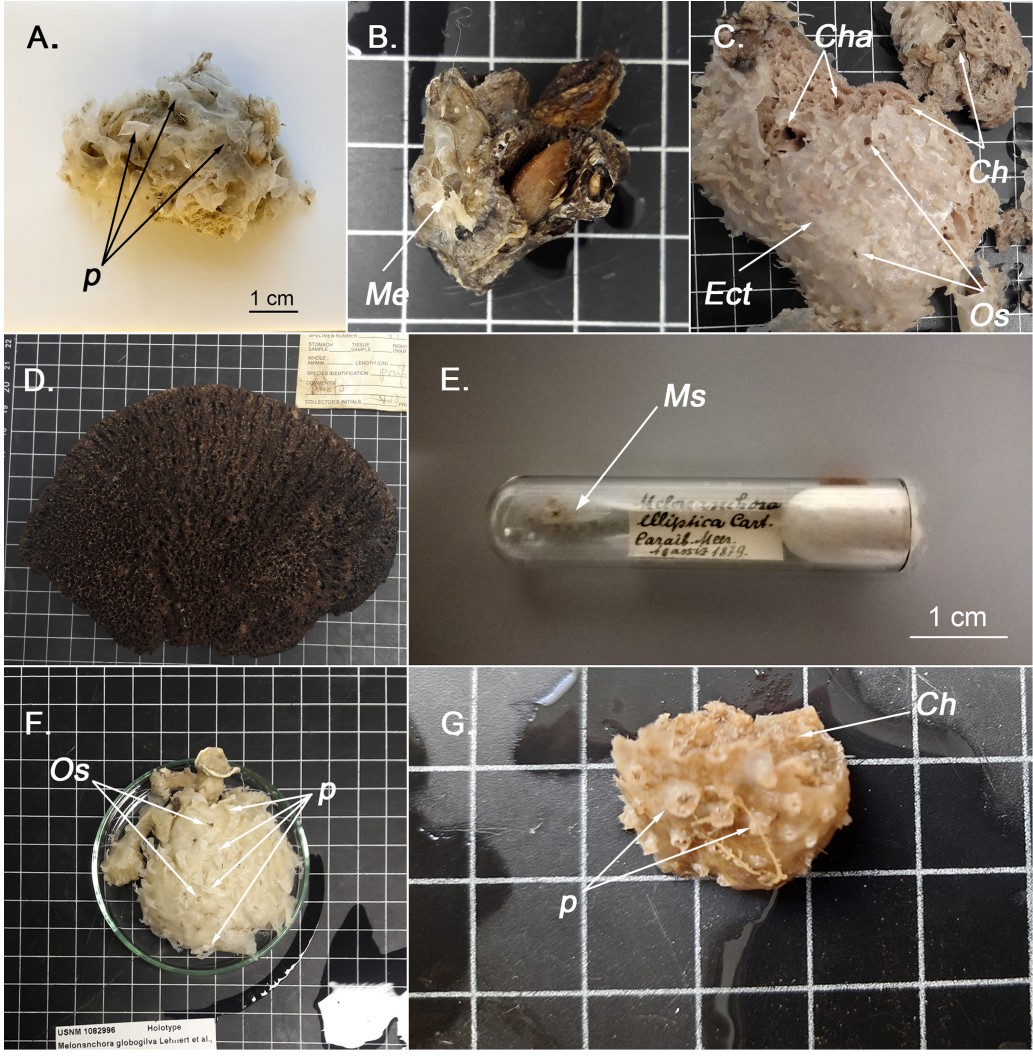

**Figure 1 External appearence of various *Melonanchora* species.** (A) External view of *Melonanchora elliptica* (MZLU L935/3858), *p* indicates some ectosomal papillae; (B) Individual of *Melonanchora emphysema* (*Me*) attached to coral rubble (GNM Porifera 416); (C) Holotype of *Melonanchora tumultuosa* sp. nov. (GNM Porifera 624), *Ect* indicates the ectosome, *Ch* indicates the choanosome, *Cha* indicates the choanosomal cavities, *Os* indicates the oscules; (D) Individual of *Arythmata tetradentifera* (NMNH-USNM 148959); (E) Holotype of *Melonanchora insulsa* sp. nov. (MZS Po165); (F) Holotype of *Hanstoreia globogilva* (NMNH-USNM 1082996), *p* indicates some ectosomal papillae and *Os* indicates the oscules; (G) Holotype of *Melonanchora maeli* sp. nov. (ZMA.POR.7269), *p* indicates some ectosomal papillae and *Ch* the choanosome.       

1885 (*Fristedt, 1887*); YPM IZ 006552.PR, Laurentian Channel, Nova Scotia, North Atlantic Ocean (44.5667, −56.6958), *USFC Albatross*, 218 m depth, 1885; NHMUK-Norman Collection 1910.1.1.588, Hardanger Fjord, ca. 180 m depth, 1882; NHMUK-Sott-Ryen Coll., 1931.6.1.19, Folden Fjord, Norway (*Burton, 1931*); NHMUK Norman Coll. 1910.1.1.1418, Norway, 1882; NHMUK–Norman Coll. 1910.1.1.1419, Norway, 1882; NHMUK–Norman Coll. 1910.1.1.1420, Norway, 1882; NHMUK–Norman Coll. 1910.1.1.1421 (*Fristedt, 1887*); NHMUK-Norwegian Coll. 1982.9.6.14.a., Norway, 1885; ZMA.POR.P.10797, North of Hammerfest, Norway, Arctic Ocean (72.15003, 22.71246),

*R/V Willem Barents* expedition (1880–84), 265 m depth, 1881 (*Vosmaer, 1885*); ZMA. POR.1548, North of Hammerfest, Norway, Arctic Ocean (72.15, 22.68333), *R/V Willem Barents* expedition (1880–84), 265 m depth, 1881 (*Vosmaer, 1885*).

Unregistered material:
NR0509_43, Flemish Cap, Tail Grand Bank, North Atlantic Ocean, 1,554 m depth (NEREIDA Coll.); NR0509_49, Flemish Cap, Tail Grand Bank, North Atlantic Ocean, 1,137 m depth (NEREIDA Coll.); NR0509_52, Flemish Cap, Tail Grand Bank, North Atlantic Ocean, 870 m depth (NEREIDA Coll.); NR0509_73, Flemish Cap, Tail Grand Bank, North Atlantic Ocean, 1,122 m depth (NEREIDA Coll.); NR0509_82a, Flemish Cap, Tail Grand Bank, North Atlantic Ocean, 1,127 m depth (NEREIDA Coll.); NR0610_21, Flemish Cap, Tail Grand Bank, North Atlantic Ocean, 1,055 m depth (NEREIDA Coll.); NR0709_5, Flemish Cap, Tail Grand Bank, North Atlantic Ocean, 1,248 m depth (NEREIDA Coll.).

Description:
Usually massive-globular sponge (Fig. 1A), more rarely encrusting (CMN 2018-0107), with an easily detachable paper-like thin ectosome bearing abundant fistular processes. The choanosome shows several scattered pores and channels. Colour whitish translucent outside, cream-orange in the choanosome.

Skeleton:
Ectosomal skeleton consists of tangential tylostrongyles with a criss-cross arrangement (Fig. 2C). Choanosomal skeleton with scattered poorly defined tracts (Fig. 2B) of styles to substyles and abundant organic content. Microscleres are distributed thorough the choanosome without any clear discernible pattern, yet, in some individuals (including the holotype), spherancorae form a dense palisade between the ectosome and the choanosome and might also cover the choanosomal tracts (Fig. 2D).

Spicular complement:
Styles, tylostrongyles, two categories of chelae, and spherancorae (Figs. 3A–3G).

Ectosomal tylostrongyles (Fig. 3B): Unevenly, slightly flexuous unequally thinning towards both ends, with a more or less central swelling and, differentially inflated ends (strongyle to tylote appearance).
Size range: 560.3–*624.3* ± 32.2–666.5 μm × 7.8–*11.8* ± 3–17.3 μm

Choanosomal styles (Fig. 3A): Entirely smooth, slightly curved towards its distal end. In general, they have the point markedly acerate, but points can also be blunt to various degrees in some spicules (stylostrongyles) (Fig. 3F).
Size range: 782.5–*830.8* ± 50–908.1 μm × 17.2–*19.3* ± 1.1–20.5 μm

Isochelae I (Fig. 3E, c'): Small anchorate isochelae, with a straight shaft, well-developed fimbriae and spatulated alae. The distal alae slightly point outwards, giving a "V" lateral appearance to both ends.

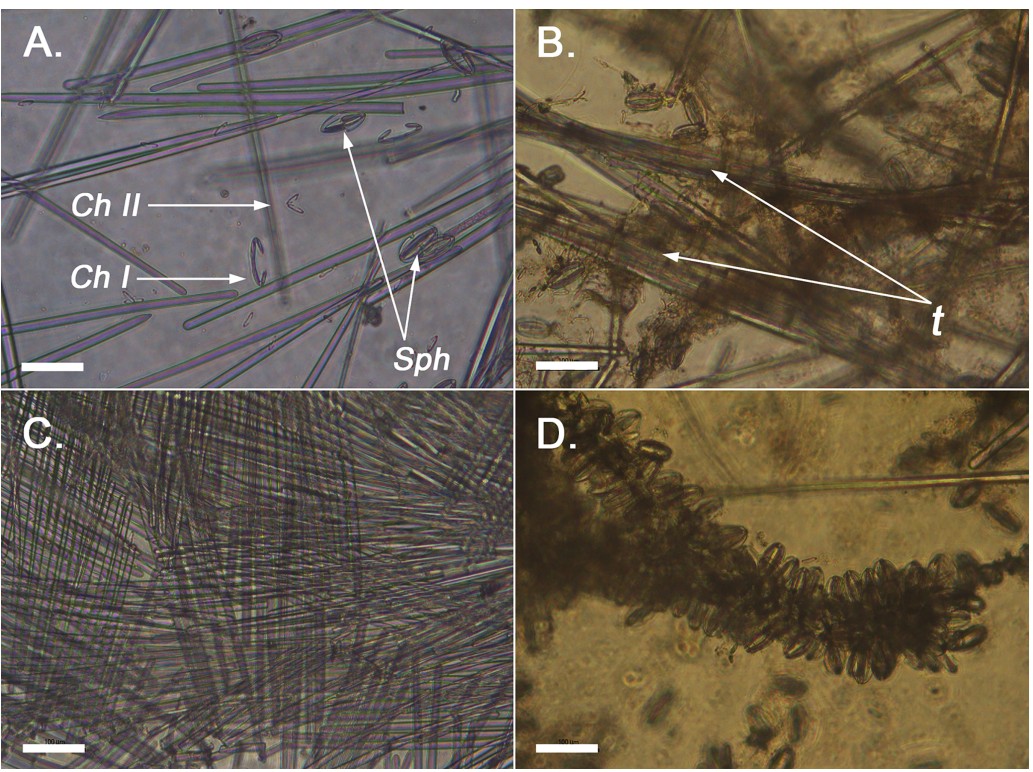

**Figure 2 Optical microscope imaging of *Melonanchora* spicules.** (A) General view of the spicules of *Melonanchora* (NHMUK 1882.7.28.54a) un light microscopy. *C. I* indicates the largest chelae category, *C. II* indicates the smallest chelae category, and *Sph* indicates spherancorae; (B) View of the loose choanosomal tracts off *Melonanchora elliptica* (NHMUK 1882.7.28.54a) (C) View of the characteristic criss-cross like pattern of the ectosome of *Melonanchora* (NHMUK–Norman Coll. 1910.1.1.1421); (D) Spherancorae covering the choanosomal tracts in *Melonanchora elliptica* (NHMUK 1882.7.28.54a).

Size range: 24.2–*26.6* ± 3.4–29 μm

Isochelae II (Fig. 3D, b'): large isochelae with a straight shaft, well-developed fimbriae and spatulated alae. The distal alae slightly point outwards, giving a "V" lateral appearance to both ends.
Size range: 48.3–*51.1* ± 3.8–58 μm

Spherancorae (Fig. 3C, a'): Unique to the genus, with an oval shape and slightly pointed ends, which might resemble a rugby ball. It possesses fimbriae on its internal face, which may be free or fused to various degrees.
Size range: 48.3–*51.2* ± 2.7–53.1 × 23.1–*28.3* ± 1.6–29.2 μm

Geographic distribution and ecological remarks:
*Melonanchora elliptica* is a common amphi-Atlantic species (Fig. 4) also occurring in Arctic waters (*Carter, 1877*), as far as the Barents Sea (*Koltun, 1959*; *Katckova et al., 2018*). It has been recorded from the coasts of Norway (*Vosmaer, 1885*; *Topsent, 1913*), Faroe Plateau (*Carter, 1874*; *Lundbeck, 1905*), Porcupine Seamount (*Könnecker & Freiwald, 2005*; *van Soest & De Voogd, 2015*) and Rockall Bank (*van Soest & Lavaleye, 2005*),

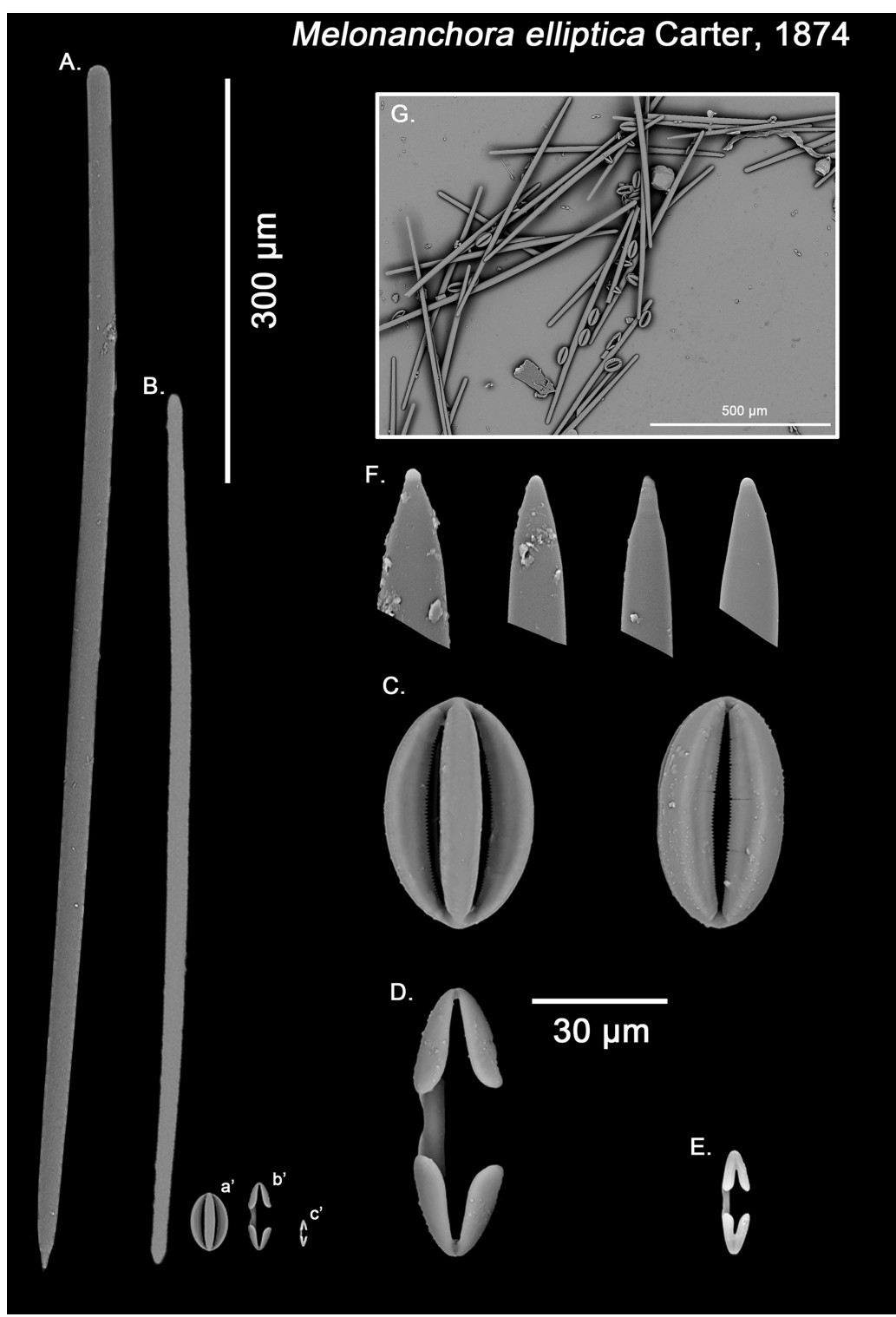

**Figure 3 *Melonanchora elliptica* spicule plate.** Spicular set for *Melonanchora elliptica* (sample NHMUK 1882.7.28.54a., holotype). (A) Choanosomal style; (B) Ectosomal tylostrongyle; (C) Spherancorae; (D) Large chelae category (Chelae II); (E) small chelae category (Chelae I); (F) Detail of the styles' acerate end; (G) General view of *M. elliptica*'s spicules by SEM imaging. (a') Spherancora
**Figure 3 (continued)**
(b') Chelae II and (c') Chelae I relative sizes when compared with that of the megascleres. Scale bars for (A), (B), (a'), (b'), (c') 300 μm; (C)–(F) 30 μm and (G) 500 μm. Images (A) to (E) and (G) were taken from sample NHMUK 1882.7.28.54a (holotype). Images for F were taken from both NHMUK 1882.7.28.54a (holotype) and CMNI 2018-0107.

Greenland and Iceland (*Lundbeck, 1905*; *Burton, 1959*), the Galician coast (*Ríos & Cristobo, 2017*), the Azores archipelago (*Topsent, 1892*, *1904*, *1928*) and the area within the Labrador Peninsula and the Newfound Land Seas (*Topsent, 1913*; *Michaud & Pelletier, 2006*; *Baker et al., 2018*), from 80 to 1,554 m depth. In the Canadian coasts and the Gulf of Maine, the species is commonly found on sponge grounds on trawlable areas (*Maciolek et al., 2008*, *2011*) and it has been observed to be an occasional nursery ground for the octopus *Rossia palpebrosa* Owen, 1935 (*Wareham Hayes, Fuller & Shea, 2017*). Nevertheless, its role and ecological significance in Vulnerable Marine Ecosystems (VMEs) are still poorly understood and in need of further research.

Remarks:

*Melonanchora elliptica* is the type species of the genus, first described from a specimen collected during the *HMS Porcupine* expedition (1869) in the Northeast Atlantic (*Carter, 1874*). The holotype description referred to a soft roundish sponge with a thin paper like ectosome with papillate projections that lodge pores and oscula. However, while the pore areas are indeed located at the wart-like papillae, the oscula are not at their tip (Figs. 1C; 1F), as initially claimed (*Carter, 1874*; *Vosmaer, 1885*) but on the ectosome (*Lundbeck, 1905*), yet they are visible only after a careful examination. The conspicuous ectosome is loosely attached to the choanosome here and there, which, together with its fragility, might contribute to its rip off during trawl sampling (*Vosmaer, 1885*; *Topsent, 1892*). Collected individuals without ectosome, appear smooth, porous, and lack the characteristic papillae. However, the presence of spherancorae facilitates the species identification, even after the ectosome's detachment (*Baker et al., 2018*).

While Carter's original description was precise, the illustrations were not sufficiently accurate. Thus, subsequent authors (*Vosmaer, 1885*; *Topsent, 1892*, *1904*) referred to Schmidt's redescription based on specimens from the Caribbean (*Schmidt, 1880*) rather than carter's description of the type specimen for their species identification. However, Schmidt's material (MZS Po165) was in fact another species (described below as *Melonanchora insulsa* sp. nov.) clearly differing from *M elliptica* in the shape of chelae and spherancorae. Finally, Topsent's individuals form the Azores are insufficiently described (Table 1) and were not available. While it is clear that they belong to *Melonanchora*, it is impossible to ascertain based on Topsent's descriptions if they unequivocally belong to *M. elliptica* or to any other North Atlantic *Melonanchora* species.

*Melonanchora emphysema* (*Schmidt, 1875*)
(Figs. 1B; 5; 6)

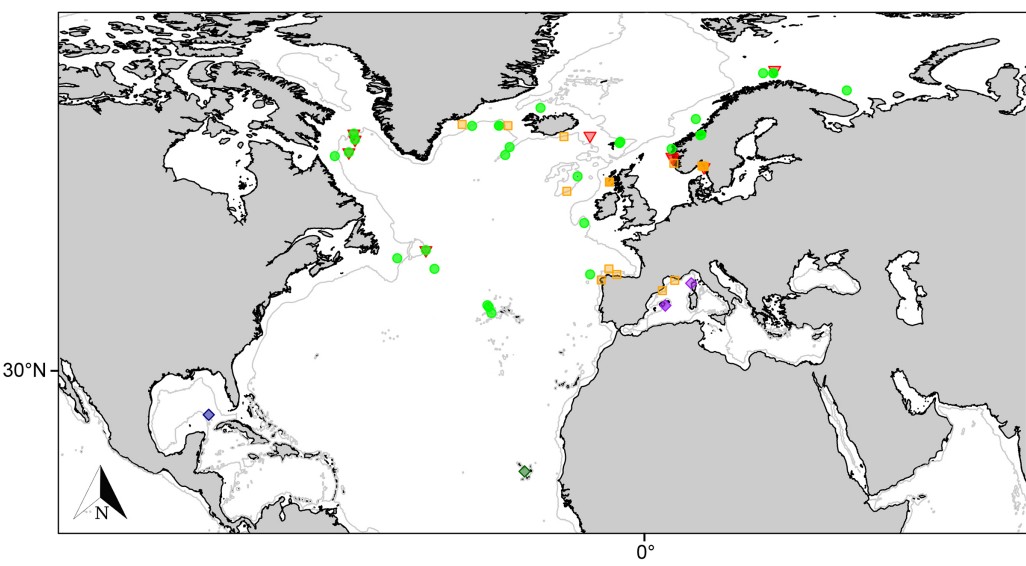

**Figure 4 Distribution map for north Atlantic *Melonanchora* species.** Distribution map for the North Atlantic *Melonanchora* species: *Melonanchora elliptica* (green circle), *Melonanchora emphysema* (orange square), *Melonanchora tumultuosa* sp. nov. (red triangle); *Melonanchora maeli* sp. nov. (dark green square); *Melonanchora intermedia* sp. nov. (purple square); *Melonanchora insulsa* sp. nov. (dark blue square). Projected view (UTM Zone 31N (WGS84)) with geographic (WGS84) coordinates indicated for reference. The 1,000 m depth isobaths is represented by a grey line. Geographic and bathymetric data used was obtained from http://www.naturalearthdata.com.

Synonymy:

*Desmacidon emphysema Schmidt, 1875*: 118.

*Melonanchora elliptica*; *Alander, 1935*: 5 (*partim*).

*Melonanchora emphysema*; *Vosmaer, 1885*: 31, pI. I fig. 14, pI. V figs. 69–70 (*partim*); *Thiele, 1903*: 393; *Lundbeck, 1905*: 213–216, pl. XX fig. 2a–2d; *Lundbeck, 1909*: 402–403; *Arndt, 1913*: 116; *Hentschel, 1929*: 966–967; *Arndt, 1935*: 73, Fig. 142; *Alander, 1942*: 57 (*partim*); *Vacelet, 1969*: 200–201, fig. 38; *Solórzano & Durán, 1982*: 105–106, fig. 5c; *Solórzano, 1990*: 755–777, L. 92; *Solórzano, 1991*: 34; *Ríos & Cristobo, 2017*: 169; *Santín et al., 2021*: Tab. 1.

Not *Melonanchora emphysema*; *van Soest, 1993*: 210, Tab. 2; *Pulitzer-Finali, 1983*: 561.

Material examined.

Holotype:

ZMB Por 2680, North Sea, from a Fjord of the southern coasts of Norway; ZMB Por 6571, North Sea, from a Fjord of the southern coasts of Norway.

Additional specimens examined:

GNM Porifera 416, Skagerrak, Sweeden, 80–100 m depth, 1934, (*Alander, 1935*, *1942*); GNM Porifera 290, Norra Kosterområdet Säcken, Baltic Sea (59.01441, 11.11977), 80 m depth, 1934, (*Alander, 1935*, *1942*); GNM Porifera 390, Norra Kosterområdet Säcken, Baltic Sea (59.01441, 11.11977), 80 m depth, 1927, (*Alander, 1935*, *1942*); MZB 2019–1740–Blanes Canyon, north-western Mediterranean Sea (41.50722, 2.93388),

**Table 1 Comparative table between all known records of *Melonanchora elliptica* *Carter, 1874*, including the locality (Loc.) and depth of the sample, as well as the measurement of their spicular complement.**

| Author | Loc./Depth | Ectosomal megascleres | Choanosomal megascleres | Isochelae | Spherancorae |
|---|---|---|---|---|---|
| *Melonanchora elliptica* *Carter, 1874* | | | | | |
| *Carter (1874)* | Faroe Plateau*/ 'deep-sea' | (St) ca. 750 μm | (S) ca. 495 μm | Present | Present |
| Reexamination *van Soest (2002)* | Faroe Plateau*/ 'deep-sea' | (St) 450–650 × 13–15 μm | (S) 650–860 μm | (I) 22–44 μm (II) 60 μm | 48–68 μm |
| Reexamination *This study* (ZMB Por 3042) | Faroe Plateau */'deep-sea' | (St) 500–*561.9* ± 34.4–*611.2* × 14.7–*15.9* ± 1.1–19.6 μm | (S) 730–*804.3* ± 78.9–1176 × 14.7–*19.2* ± 2.1–22.2 μm | (I) 22.8–*25* ± 1.5–27.6 μm (II) 48.9– *61* ± 2.4–66.3 μm | 58.8–*62.4* ± 2.2–68.3 × 27.6–*29.7* ± 1.8–31.3 μm |
| *Vosmaer (1885)* | Barents Sea | Present | Present | Present | Present |
| Reexamination *This study* (ZMA.POR. P.10797) | Barents Sea | (St) 584–*678* ± 55.9–762 × 13.8–*16.8* ± 1.7–18.6 μm | (S) 738–*994.3* ± 89.9–1146 x 15–*19.1* ± 2.7–23.7 μm | (I) 24–*27.8* ± 1.5–31 μm (II) 63–*71.8* ± 2.3–81 μm | 63–*67.5* ± 2.2–72 × 26–*28.9* ± 1.7–30.5 μm |
| *Fristedt (1887)* | East Greenland/ 580 m | (St) 500 μm | *nm* | (I) 15 μm (II) 60 μm | 70 μm |
| *Arnesen (1903)* | Between Bergen and Trondheim/ 100–180 m | *nm* | (S) ca. 1000 μm | (I) *nm* (II) 68 μm | 60 μm |
| *Lundbeck (1905)* | North Atlantic/ 105–1,460 m | (St) 410–620 × 8–17 μm | (S) 680–860 × 14–21 μm | (I) 21–28 μm (II) 47–61 μm | 54–68 × 24–38 μm |
| *Arndt (1935)* | North Atlantic/ 'deep-sea' | (St) 410–620 μm | (S) 680–860 μm | (I) 21–28 μm (II) 47–75 μm | 54–68 μm |
| *Koltun (1959)* | Barents Sea/ 106–385 m | (St) 410–620 × 8–17 μm | (S) 680–904 × 14–27 μm | *nm* | *nm* |
| *Baker et al. (2018)* | Davis Strait/ 537–1,132 m | (St) 528.1–*594.7*–655.5 × 14.2–*19.3*–23.9 μm | (S) 689.7–*842.8*–902.8 × 11.1–*15.1*–21.1 μm | (I) 23.1–*25.4*–28.8 μm (II) 40.4–*57.4*–67.6 μm | 48–*57.2*–65.7 × 24–*29.7*–35.9 μm |
| | | (St) 575.9–*618.6*–661.5 × 18.3–*21.6*– 24.8 μm | (S) 730.2–*778.4*–822.4 × 13.3–*15.5*–17.9 μm | (I) 22.7–*24.9*–27 μm (II) 44.7–*54.8*–61.6 μm | 54.1–*62.8*–68 × 26.9–*31*–36.9 μm |
| | | (St) 497.4–*613.1*–725.5 × 15.7–*19.5*–22.2 μm | (S) 701.8–*759.8*–827.4 × 12–*14.5*–19 μm | (I) 21.4–*25.1*–29.1 μm (II) 50.9–*56.9*–60.8 μm | 51.2–*57.9*–63.4 × 23.7–*30.1*–37.5 μm |
| | | (St) 504.4–*568*–629.1 × 16–*19.2*–22.7 μm | (S) 743.5–*814.3*–879.1 × 11.3–*14.4*–18.8 μm | (I) 23.2–*26*–27.2 μm (II) 48.2–*52.5*–57.7 μm | 46.3–*55.8*–61.7 × 25.6–*29*–33.2 μm |
| | | (St) 498.4–*553*–603 × 15.7–*18.6*–22.3 μm | (S) 682.2–*758.4*–835.4 × 13.5–*17.4*–20.5 μm | (I) 21.5–*24.4*–26.3 μm (II) 42.1–*59*–82.8 μm | 41.5–*49.5*–57.5 × 27.8–*31.8*–37.9 μm |
| *Dinn & Leys (2018)* | Saglek Bank, Northern Labrador Sea/ 427 m | (T) 554–*623*–693 × 12.6–*15.5*–18.6 μm. | (S) 749–*833*–923 × 18.5–*23*–26 μm | (I) 18–*22*–27.6 μm (II) 35–*55*–64 μm | 43–*50*–53 μm |
| Reexamination *This study* (CMNI 2018-0107) | Saglek Bank, Northern Labrador Sea/ 427 m | (St) 560.3–*624.3* ± 32.2–*667.6* × 7.8–*11.8* ± 3–17.3 μm. | (S) 782.5–*830.7* ± 50–908 × 19.3–*21.5* ± 1.2–23.1 μm | (I) 24.1–*24.9* ± 1.2–29 μm (II) 48.3–*51* ± 3.8–59 μm | 48.3–*51.2* ± 2.6–53.1 × 26.5–*29* ± 0.7–29.8 μm |
| *This study* (NR0509_43) | Flemish Cap, Tail Grand Bank/ 1,554 m | (St) 533–645 × 6–13 μm | (S) 619–803 × 14–18 μm | (I) 21–26 μm (II) 46–66 μm | 48–64 × 20–33 μm |

(Continued)

| Author | Loc./Depth | Ectosomal megascleres | Choanosomal megascleres | Isochelae | Spherancorae |
|---|---|---|---|---|---|
| **Table 1 (continued)** | | | | | |
| *This study* (NR0509_49) | Flemish Cap, Tail Grand Bank/ 1,137 m | (St) 488–610 × 8–17 μm | (S) 601–1000 × 15–27 μm | (I) 20–30 μm (II) 50–67 μm | 52–61 × 19–28 μm |
| *This study* (NR0509_52) | Flemish Cap, Tail Grand Bank/ 1,122 m | (St) 504–598 × 12–16 μm | (S) 751–1086 × 16–24 μm | (I) 21–35 μm (II) 55–77 μm | 55–66 × 26–39 μm |
| *This study* (NR0509_73) | Flemish Cap, Tail Grand Bank/870 m | (St) 555–625 × 11–17 μm | (S) 767–910 × 15–24 μm | (I) 25–29 μm (II) 39–70 μm | 51–63 × 23–34 μm |
| *This study* (NR0509_82a) | Flemish Cap, Tail Grand Bank/ 1,127 m | (St) 538–676 × 12–20 μm | (S) 637–867 × 17–20 μm | (I) 22–28 μm (II) 51–71 μm | 58–68 × 27–39 μm |
| *This study* (NR0620_21) | Flemish Cap, Tail Grand Bank/ 1,248 m | (St) 532–842 × 10–19 μm | (S) 722–902 × 10–22 μm | (I) 19–27 μm (II) 38–52 μm | 46–59 × 25–35 μm |
| *This study* (NR0709_5) | Flemish Cap, Tail Grand Bank/ 1,055 m | (St) 518–845 × 11–20 μm | (S) 705–833 × 13–22 μm | (I) 23–33 μm (II) 37–63 μm | 50–62 × 26–35 μm |
| *This study* (NHMUK Norman Coll. 1910.1.1.1418) | Norway/unknown | (St) 479.5–*602.8* ± 24.1–673 x 14.3–*16.4* ± 2.2–19.1 μm | (S) 765–*863.8* ± 59.5–925.7 x 15.3–*19.8* ± 1.5–21.7 μm | (I) 24.3–*27.1* ± 2.4–33.3 μm (II) 61–*72.6* ± 8–82 μm | 67–*75.6* ± 5.4–82.6 × 27.1–*31.7* ± 4.3–35.4 μm |
| *This study* (NHMUK Norman Coll. 1910.1.1.1419) | Norway/unknown | (St) 548–*570.3* ± 10.3–628 x 13.7–*15.8* ± 1.8–18.7 μm | (S) 745.6–*880.1* ± 34.9–936 x 14.9–*18.5* ± 1.3–23.5 μm | (I) 26–*27.2* ± 0.8–28.5 μm (II) 67.3–*75.5* ± 1.4–78 μm | 67–*75.2* ± 6.5– 83 × 23.7–*33.1* ± 6.5–36 μm |
| ***Melonanchora* cf. *elliptica* Carter, 1874** | | | | | |
| *Topsent (1892)* | Azores/736–1,267 m | (St) Present | (S) Present | (I) *nm* (II) 55 μm | 70 μm |
| *Topsent (1904)* | Azores/523–1,360 m | *nm* | *nm* | (I) 18–21 μm (II) *nm* | *nm* |
| *Topsent (1913)* | Norwegian coast/ 440 m | *nm* | *nm* | *nm* | *nm* |
| *Topsent (1928)* | Azores/650–950 m | *nm* | *nm* | (I) 19–23 μm (II) 40–41 μm | 43 × 26 μm |
| | Azores/1,378 m | *nm* | *nm* | (I) 20–23 μm (II) 72 μm | 72 × 35 μm |

**Notes:**
(S) indicates styles; (St) indicates strongyles; (T); indicates tylostyles.
* indicates this is the holotype of the species; *nm* indicates a spicular type that was not mentioned on a description, yet it is assumed was present on the samples.

'ABIDES' survey, 684 m depth, 2018 (*Santín et al., 2021*); ZMA.POR.P.10800 Outer Hebrides, Scotland, North-East Atlantic (56.80588, −7.42903), 2006; ZMA.POR.20192 Outer Hebrides, Scotland, North-East Atlantic (56.80588, −7.42903), 2006; ZMA.POR. P.10799 West of Hvasser, Norway, Baltic Sea (59° 04′ 42.06″N 10° 43′ 55.379″E), 2006; ZMA.POR.20559.b West of Hvasser, Norway, Baltic Sea (59.07835, −10.73204), 2006; ZMA.POR.20473.b West of Hvasser, Norway, Baltic Sea (59.07835, −10.73204), 2006; ZMA.POR.20551 West of Hvasser, Norway, Baltic Sea (59.07835, −10.73204), 2006; ZMA.

POR.P.10798 Outer Hebrides, Scotland, North-East Atlantic (56.8071, −7.43025), 2006; ZMA.POR.20353.a Outer Hebrides, Scotland, North-East Atlantic (56.8071, −7.43025), 2006; ZMA.POR.P.10795 West of Ireland, North-East Atlantic (55.50093, −15.78839), attached to *Madrepora* debris, 2005; ZMA.POR.P.20020 West of Ireland, North-East Atlantic (55.50093, −15.78839), attached to *Madrepora* debris, 2005; ZMA.POR.20020 West of Ireland, North-East Atlantic (55.50093, −15.78839), attached to *Madrepora* debris, 2005; ZMA.POR.P.10829 West of Hvasser, Norway, Baltic Sea (59.07577, 10.73552), 2007; ZMA.POR.20467 West of Hvasser, Norway, Baltic Sea (59.07577, 10.73552), 2007; ZMA.POR.P.10828 Outer Hebrides, Scotland, North-East Atlantic (56.8059, −7.44183), 2006; ZMA.POR.20175.b Outer Hebrides, Scotland, North-East Atlantic (56.8059, −7.44183), 2006; ZMA.POR.P.10827 Outer Hebrides, Scotland, North-East Atlantic (56.80563, −7.426029), 2006; ZMA.POR.20335 Outer Hebrides, Scotland, North-East Atlantic (56.80563, −7.426029), 2006.

Unregistered material:
AVILES_0710–48DR5, Avilés Canyon System, Cantabrian Sea (43.80333, −6.15583), 128 m depth (INTEMARES AVILES Coll.); MS, off Bares (44.055, −7.64638), Spanish coasts, 500 m depth; JV, Cassidaigne Canyon (42.95, 5.38333), 360 m depth (*Vacelet, 1969*); Galician Bank, west of Galician coast, Spain (42.58305, −11.58305) ca. 700 m depth; Baixo do Placer do Cabezo de Laxe (43, −9.03333), Galicia Coast, Spain, Fishermen's by-catch, 58 m depth, 1981 (*Duran & Solórzano, 1982*; *Solórzano, 1990*, *1991*).

Description:
Mostly encrusting, rarely massive-encrusting (GNM Porifera 416), with an easily detachable paper-like ectosome bearing fistular processes. Fistulae might be absent in small encrusting individuals. Colour whitish translucent in the ectosome, cream-orange in the choanosome while in alcohol.

Skeleton:
Ectosomal skeleton formed by intertwined tangential tylostrongyles. The choanosomal skeleton is ill defined, with scattered tracts of tylostrongyles identical to those conforming the ectosome. Microscleres mostly scattered thorough the choanosome without any clear discernible pattern.

Spicule complement:
Tylostrongyles, two categories of chelae, and spherancorae (Figs. 5A–5E and Figs. 6A–6F).

Ectosomal and choanosomal tylostrongyles (Figs. 5A; 6A): of similar shape to those of *M. elliptica*: they are unevenly and slightly flexuous, enlarged at the central zone and narrowing toward unequal tylotoid (Fig. 6F), giving them the appearance from strongyles to tylostrongyles.
Size range: 492.7–*508.1* ± 13–521.6 μm × 9.7–*10.6* ± 2.8–14.5 μm

Isochelae I (Figs. 5D, c'; 6E, c'): Small isochelae with a straight shaft, gently bending to its ends, with three spatulated alae and well-formed fimbriae.

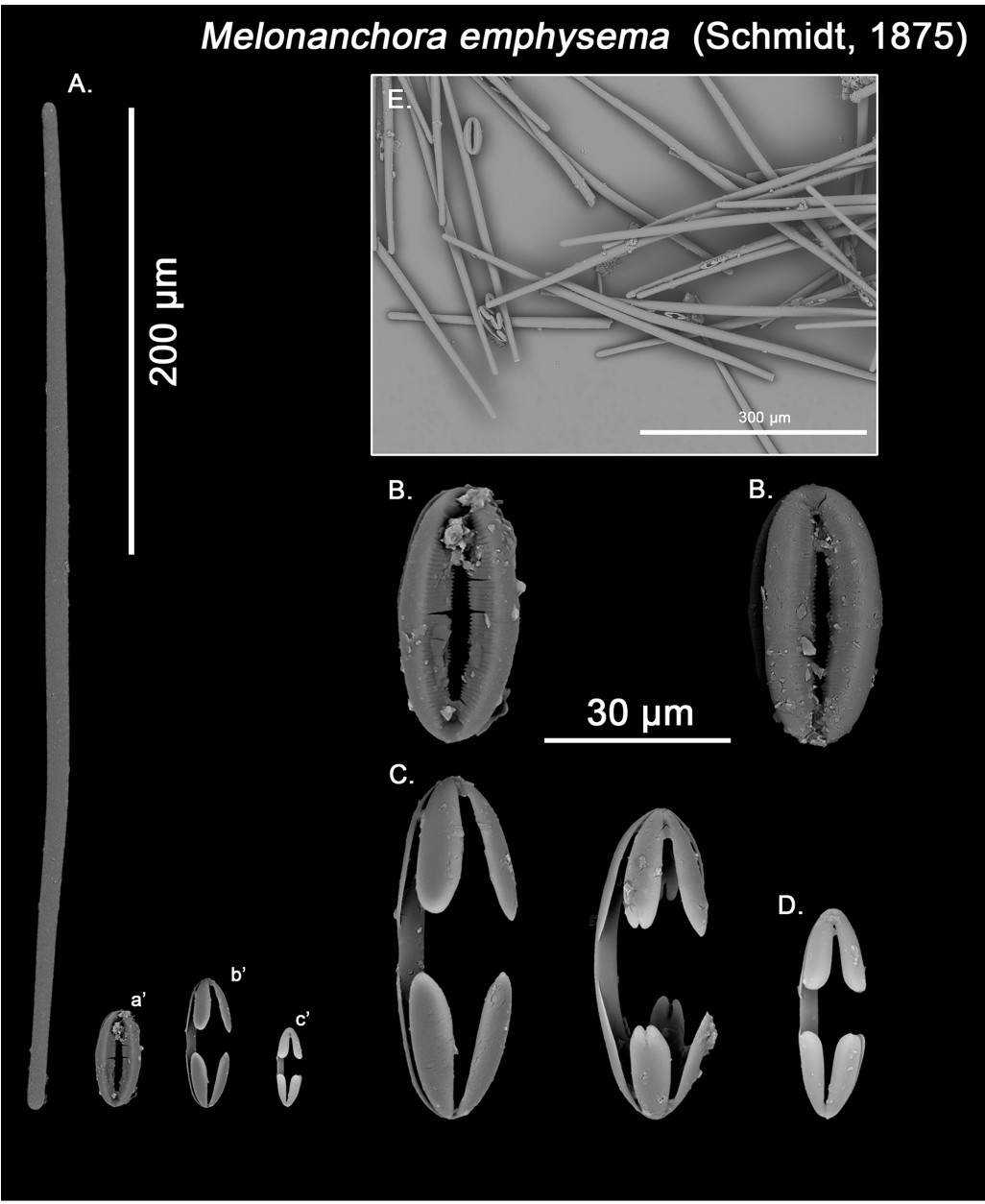

**Figure 5** *Melonanchora emphysema* **spicule plate.** Spicular set for *Melonanchora emphysema* (sample ZMB Por 2680, holotype). (A) Ectosomal and chonasomoal tylostrongyle; (B) Spherancorae; (C) Large chelae category (Chelae II); (D) small chelae category (Chelae I); (E) General view of *M. emphysema*'s spicules by SEM imaging. (a') Spherancora (b') Chelae II and (c') Chelae I relative sizes when compared with that of the megascleres. Scale bars for (A), (a'), (b'), (c') 200 µm; (B), (C), (D) 30 µm and (E) 500 µm.

Size range: 24.1–*26.6* ± 2.8–28.9 µm

Isochelae II (Figs. 5C, b'; 6C, b'): very similar to isochelae I, but bigger in size.
Size range: 48.3–*51.5* ± 5.5–58 µm

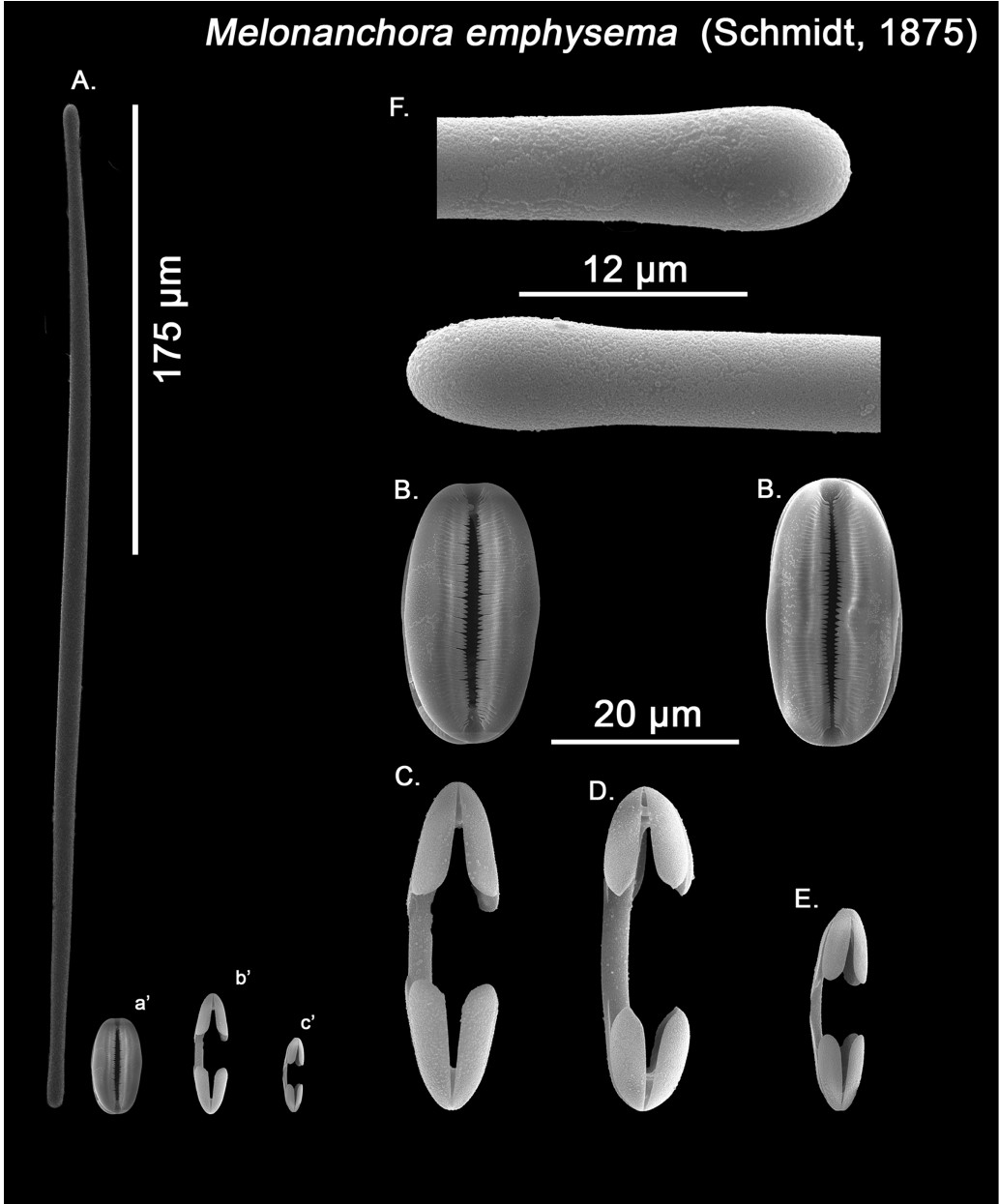

**Figure 6** *Melonanchora* **cf. *emphysema* spicule plate.** Spicular set for *Melonanchora* cf. *emphysema* from Laxe, Galicia coast, Spain (unregistered sample). (A) Ectosomal and chonasomoal tylostrongyle; (B) Spherancorae; (C) Large chelae category (Chelae II); (D) Chelae II with reduced alae; (E) small chelae category (Chelae I); (F) Detail of the tyles. (a') Spherancora (b') Chelae II and (c') Chelae I relative sizes when compared with that of the megascleres. Scale bars for (A), (a'), (b'), (c') 175 µm; (B), (C), (D) 20 µm and (F) 12 µm.

Spherancorae (Figs. 5B, a'; 6B, a'): Elongated-ovoid (Fig. 5B) to stadium shaped (Fig. 6B) with teeth-like fimbriae on its internal surface, which may be fused at various degrees. Size range: 37.6–*38.8* ± 1.1–40.5 × 25.1–*27.6* ± 1.6–28.9 µm

Geographic distribution:
Originally described from the coasts of Norway (*Schmidt, 1875*), the species is known from deep Atlantic and Arctic waters (Fig. 4), including Greenland and Iceland, (*Lundbeck, 1905*, *1909*, *1910*), Faroe Islands (*Hentschel, 1929*), Porcupine Bank (*van Soest & De Voogd, 2015*), Baltic Sea (*Alander, 1935*, *1942*), the Spanish coasts (*Solórzano, 1990*; *Ríos & Cristobo, 2017*; this paper), and the coasts of Norway (*Vosmaer, 1885*; *Arndt, 1913*) including the Svalbard archipelago (*Gulliksen et al., 1999*). The species had also been tentatively recorded from the Atlantic Canadian coast (*Baker et al., 2018*; *Murillo et al., 2018*), yet these records correspond to *Melonanchora tumultuosa* sp. nov., thus its presence in the west Atlantic area remaining unconfirmed. Additionally, the species has also been sparsely recorded from the Mediterranean Sea and nearby areas: the Gulf of Lyon (*Vacelet, 1969*; *Santín et al., 2021*) and the northern coasts of Spain (*Solórzano & Durán, 1982*; *Solórzano, 1990*, *1991*; *Ríos & Cristobo, 2017*; this study). The species appears to be a frequent inhabitant of cold-water corals communities (*Könnecker & Freiwald, 2005*; *van Soest & De Voogd, 2015*), yet it might also occur attached to rocky substrata or debris.

Remarks:
*Schmidt (1875)* poorly described *Desmacidon emphysema* from the coast of Norway, a species characterized by the presence of a papillate ectosome and smooth megascleres enlarged at the middle, with unequally swelled ends. While Schmidt accurately reported spherancorae in his *M. emphysema* samples from the Caribbean (*Schmidt, 1880*), he missed these spicules in the Northern Sea samples, mistaking them with diatoms (*Schmidt, 1875*), which led to his misclassification of *M. emphysema* in the genus *Desmacidon*, until amended by *Thiele (1903)*. Furthermore, Schmidt's incomplete description (Table 2) led several authors to consider the species a synonym of *M. elliptica* (*Vosmaer, 1885*; *Arnesen, 1903*) while others claimed that a clear distinction existed (*Thiele, 1903*; *Lundbeck, 1905*). The problem mainly arose as the main distinguishing feature between both species relies on its choanosomal megascleres, with *M. elliptica* possessing styles and *M. emphysema* possessing strongyles (*Lundbeck, 1905*), yet several authors had described samples with blunt-ended styles as choanosomal megascleres (*Vosmaer, 1885*; *Baker et al., 2018*).

The re-examination of Schmidt holotype (ZMB Por 2680) however leaves no doubt about the validity of the species. As previously pointed out (*Thiele, 1903*; *Lundbeck, 1905*), *M. emphysema*'s choanosomal megascleres are exclusively tylostrongyles identical to its ectosomal ones, while its spherancorae are smaller or equal in size to the large isochelae (Table 2). Conversely, in *M. elliptica* there is a clear distinction between the choanosomal (styles) and ectosomal (tylostrongyles) megascleres and, additionally, the spherancorae are within the size range of the large isochelae (Table 1). Thus, individuals identified as *M. emphysema* with blunt-ended diactines in two clear categories do not correspond to this species but to a new one, *Melonanchora tumultuosa* sp. nov. (described below). Finally, in the Mediterranean and nearby areas, *M. emphysema* tylostrongyles are almost half in size than those in the North Atlantic specimens (average length ca. 400 *vs.* 600 μm; Table 2), and it has been suggested that they might correspond to a yet

**Table 2 Comparative table between all known records of *Melonanchora emphysema* (*Schmidt, 1875*), including the locality (Loc.) and depth of the sample, as well as the measurement of their spicular complement.**

| Author | Loc./Depth | Ectosomal megascleres | Choanosomal megascleres | Isochelae | Spherancorae |
|---|---|---|---|---|---|
| *Melonanchora emphysema* (*Schmidt, 1875*) | | | | | |
| *Schmidt (1875)* | Haugesund, Norway*/ 193 m | **(St)** Present | *nm* | Present | *nm* |
| Reexamination *This study* (ZMB Por 2680) | Haugesund, Norway*/ 193 m | **(St)** 500–*570* ± 15.9–627 × 10.9–*15.8* ± 3.1–18.5 μm | Same as in ectosome | **(I)** 19.6–*24.7* ± 2.7–29.4 μm **(II)** 55.3–*60.2* ± 3.9–68.6 μm | 40.4–*44.3* ± 1.8–58 × 23.1–*25.6* ± 1.3–28 μm |
| *Thiele (1903)* | North Atlantic | **(St)** ca. 650 μm | Same as in ectosome | **(I)** 21 μm **(II)** 60 μm | 50 μm |
| *Lundbeck (1905)* | North Atlantic/ 375–1,460 m | **(St)** 440–610 × 10–14 μm | Same as in ectosome | **(I)** 24–30 μm **(II)** 57–71 μm | 50–56 × 28 μm |
| *Alander (1942)* | Skandia, Sweden/85 m | Present | Present | Present | Present |
| Reexamination *This study* (GNM Porifera 390) | Skandia, Sweden/85 m | **(St)** 492.7–*508.1* ± 13–521.7 × 9.7–*10.6* ± 2.8–14.5 μm | Same as in ectosome | **(I)** 24.2–*26.6* ± 2.7–29 μm **(II)** 48.3–*51.5* ± 5.5–58 μm | 37.6–*38.9* ± 1–42.6 × 21.6–*24.3* ± 1.6–29 μm |
| *Vacelet (1969)* | Mediterranean/ 360–370 m | **(St)** 330–490 × 8.5–18 μm | Same as in ectosome | **(I)** 22 μm **(II)** 40–53 μm | 40–45 × 20 μm |
| Rexamination *This study* (unregistered) | Mediterranean/ 360–370 m | **(T)** 389.3–*418.6* ± 11.7–477 × 12.2–*14.6* ± 1.3–17.6 μm | Same as in ectosome | **(I)** 21.4–*22.9* ± 0.9–25.3 μm **(II)** 41.2–*45* ± 1.2–55.1 μm | 38.4–*41.3* ± 1.5–44.5 × 17.1–*19.7* ± 2.3–22.7 μm |
| *This study* (ZMA.POR.P.10800) | Scotland/- | **(St)** 342–*472.8* ± 61.8–540 × 5.4–*6.9* ± 0.8–7.8 μm | Same as in ectosome | **(I)** 22.8–*24.3* ± 1–25.8 μm **(II)** 48–*52.5* ± 5.6–63 μm | 37.8–*41.7* ± 2.8–44.4 × 18–*19.5* ± 1.3–21 μm |
| *This study* (unregistered) | Galicia Bank/ 500 m | **(T)** 439.2–*479.9* ± 30.4–537.6 × 12.2–*15.5* ± 1.8–18.7 μm | Same as in ectosome | **(I)** 20.7–*23.4* ± 1.5–25.4 μm **(II)** 42–*51.2* ± 4.3–57.2 μm | 37.2–*41.2* ± 2–44.6 × 17.3–*20.6* ± 1.2–23.4 μm |
| *This study* (unregistered) | Galicia Bank/ 500 m | **(T)** 429.2–*482.2* ± 29.7–538.9 × 11.8–*15* ± 1.7–18.7 μm | Same as in ectosome | **(I)** 20.2–*22.8* ± 1.9–27.3 μm **(II)** 40.6–*54* ± 4.8–62.7 μm | 34.7–*41.2* ± 4–54.5 × 17.2–*20.2* ± 2–23.5 μm |
| *This study* (MZB 2019–1740) | Gulf of Lyon/ 684 m | **(T)** 253.6–*375.6* ± 48.7–426.1 μm × 8.8–*10.1* ± 1.7–13.7 μm | Same as in ectosome | **(I)** 20.5–*24.1* ± 3.7–30.4 μm **(II)** 44.3–*53* ± 4.2–60 μm | 41.2–*43.7* ± 2.1–46.6 × 18.3–*20.5* ± 2.7–26.3 μm |
| *Melonanchora* cf. *emphysema* (*Schmidt, 1875*) | | | | | |
| Solórzano & Duran (1981) | Galicia Coast, Spain*/58 m | **(St)** 316–345 × 9 μm | Same as in ectosome | **(I)** 22– 26 μm **(II)** 44 –51 μm | 27–40 μm |
| Reexamination *Solórzano (1990)* | Galicia Coast, Spain*/58 m | **(St)** 316–345 × 8–9 μm | Same as in ectosome | **(I)** 22– 26 μm **(II)** 44 –51 μm | 27–40 × 18–20 μm |

(Continued)

| Author | Loc./Depth | Ectosomal megascleres | Choanosomal megascleres | Isochelae | Spherancorae |
|---|---|---|---|---|---|
| Reexamination *This study* (unregistered) | Galicia Coast, Spain*/58 m | **(T)** 302.6–*345.8* ± 24–384.5 × 4.9–*6.83* ± 0.8–8 μm | Same as in ectosome | **(I)** 16.5–*20* ± 1.4–22.2 μm **(II)** 35–*44* ± 3.9–50 μm | 31.9–*36.2* ± 2.3–40.5 × 14.2–*17.2* ± 2.1–20.5 μm |
| *This study* (AVILES_0710–48DR5) | Cantabrian Sea/ 128 m | **(T)** 274–*329.6* ± 30.6–387.6 × 4.6–*6.1* ± 0.8–7.6 μm | Same as in ectosome | **(I)** 15.4–*18* ± 1.3–20.7 μm **(II)** 33.6–*44* ± 3.8–48.9 μm | 34.7–*37.2* ± 1.2–39.3 × 12.6–*16* ± 2–19.9 μm |

Notes:
**(S)** indicates styles; **(St)** indicates strongyles; **(T)**; indicates tylostyles.
* indicates this is the holotype of the species; *nm* indicates a spicular type that was not mentioned on a description, yet it is assumed was present on the sample/s.

undescribed species (*Vacelet, 1969*). In this sense, reexamination of all known Mediterranean material did in fact reveal a new species, *Melonanchora intermedia* sp. nov. (described below), occurring within western Mediterranean meshophotic environments (*Pulitzer-Finali, 1983*; *Díaz, Ramírez-Amaro & Ordines, 2021*). However, no major differences could be observed with *M. emphysema* specimens from other deep-sea Mediterranean and nearby areas other than the aforementioned size of their tylostrongyles (Table 2). Additionally, the Mediterranean and Iberian specimens' spherancorae (Fig. 6B) closely match a stadium-shaped appearance, which is characteristic of *M. emphysema*. However, it must be noted that one specimen from the Galician coast and another one from the Cantabrian Sea possess relatively smaller and thinner tylostrongyles (ca. 330 μm length *vs.* ca. 6 μm width) when compared with all other *M. emphysema* records (Table 2), and, in the Galician sample, an additional category of chelae with reduced alae could be observed in very low numbers (Fig. 6D). Nevertheless, said chelae are absent from all other Iberian or Mediterranean *M. emphysema* material. Given the high variability in megasclere size observed within all *Melonanchora* species (Tables 1–3), as well as the poor conservation status of these deviant samples, it would be unwise to erect a new species based solely on the megascleres size. Yet, the possibility that those specimens correspond in fact to a cryptic species cannot be entirely ruled out, and its identity should be further clarified if more individuals with said characteristics were to be discovered.

*Melonanchora tumultuosa* sp. nov.
(Figs. 1C; 7)

Synonymy:
*Melonanchora elliptica*; *Vosmaer, 1885*: 31, pI. I fig. 14, pI. V figs. 69–70 (*partim*); *Lundbeck, 1905*: 213–216, pl. VII figs. 4–6, pl. XX figs. 1a–1o (*partim*); *Lundbeck, 1909*: 402–403 (*partim*); *Alander, 1935*: 5 (*partim*).
*Melonanchora emphysema*; *Alander, 1942*: 57 (*partim*); *Baker et al., 2018*: 26–30, figs. 8–10. Not *Melonanchora elliptica Carter, 1874*: 212.

Material examined.

**Table 3 Comparative table for all new species of *Melonanchora*, as well as the closely related genus *Hanstoreia* gen. nov., described on this work, including: the locality (Loc.) and depth of the sample, as well as the measurement of their spicular complement.**

| Author | Loc./Depth | Ectosomal megascleres | Choanosomal megascleres | Isochelae | Spherancorae |
|---|---|---|---|---|---|
| ***Melonanchora tumultuosa* sp. nov.** | | | | | |
| *Vosmaer (1885)* | – | Present | Present | Present | Present |
| Reexamination *This study* (ZMA.POR. P.10796) | Norway/256 m | (St) 483–*542.6* ± 38.3–600 μm × 10.6–*12.9* ± 3.2–19.3 μm | (St) 627.9–*802.3* ± 42.2–924.5 μm × 11.6–*18.3* ± 1.5–24.4 μm | (I) 21.2–*26.5* ± 3.8–28.9 μm (II) 48.6–*68.6* ± 8.1–72.9 μm | 48.3–*67.5* ± 6.8–78.62 × 18.9–*22.3* ± 1.6–25.2 μm |
| *Baker et al. (2018)* | Davis Strait/ 537–1132 m | (St) 485.1–*599.8*–673.3 × 12.7–*15.6*–20 μm | (St) 831.1–*913.6*–981.6 × 15.7–*19.5*–22.7 μm | (I) 22.6–*25.8*–32.2 μm (II) 43.3–*59*–66.4 μm | 53.2–*57.5*–63.7 × 23.1–*27.7*–35.3 μm |
| | | (St) 537.5–*582.6*–670.8 × 12.0–*14.4*–17.4 μm | (St) 823.5–*884.6*–957.8 × 13.5–*19.2*–24 μm | (I) 22.2–*24.3*–27.1 μm (II) 44–*49.5*–56.8 μm | 52.8–*54.9*–59.3 × 24.9–*30.4*–36.0 μm |
| | | (St) 509.9–*569.8*–611.6 × 11.3–*14.7*–17.9 μm | (St) 672.6–*770.9*–860.1 × 17.4–*20*–23.9 μm | (I) 20.5–*22.7*–25.4 μm (II) 49.5–*52.3*–56.3 μm | 57.5–*61.7*–65.1 × 23.9–*26.9*–28.8 μm |
| *This study* (NR0509_82b) | Flemish Cap, Tail Grand Bank/ 1,027 m | (St) 548–*657* × 11–17 μm | (St) 716–*873* × 14–22 μm | (I) 22–26 μm (II) 49–68 μm | 56–67 × 25–38 μm |
| *This study* (NR0610_30) | Flemish Cap, Tail Grand Bank/613 m | (St) 544–*657* × 8–18 μm | (St) 483–*823* × 8–13 μm | (I) 24–32 μm (II) 38–67 μm | 47–65 × 22–34 μm |
| *This study* (GNM Porifera 624) | Sydkoster Island, Sweeden*/100 m. | (St) 483–*542.6* ± 38.3–600 × 10.6–*12.9* ± 3.2–19.3 μm | (St) 627.9–*802.3* ± 42.2–924.5 × 11.6–*18.3* ± 1.5–24.4 μm | (I) 21.2–*26.5* ± 3.8–28.9 μm (II) 48.6–*68.6* ± 8.1–72.9 μm | 48.3–*67.5* ± 6.8–78.6 × 18.9–*22.3* ± 1.6–25.2 μm |
| *This study* (NHMUK, 83.12.13.70.89) | Unknown | (St) 483–*542.6* ± 38.3–600 × 10.6–*12.9* ± 3.2–19.3 μm | (St) 768–*895.7* ± 38.3–993 × 15.7–*19.8* ± 1.6–24 μm | (I) 18.5–*21* ± 2.6–25 μm (II) 55.7–*76.1* ± 2.9–79 μm | 62.8–*70* ± 4.9–78 × 22.1–*24.5* ± 1.9–29.3 μm |
| *This study* (NHMUK Norman Coll. 1898.5.7.38) | Norway | (St) 490–*550.4* ± 38.9–607.6 × 10.8–*13.1* ± 3.3–19.6 μm | (St) 637–*712.7* ± 31.3–813.5 × 11.8–*14.7* ± 1.5–21.1 μm | (I) 21.3–*26.5* ± 2.5–29 μm (II) 40.2–*57.7* ± 8.2–69.6 μm | 48.3–*60* ± 4.2–67.6 × 25.1–*27* ± 1.5–29 μm |
| *This study* (ZMA.POR. P.10825) | Norway/130–150 m | (St) 528–*617* ± 52.2–667 × 12.8–*15* ± 2–18 μm | (St) 642–*696* ± 58.8–804.3 × 14.7–*18.6* ± 2.7–21.9 μm | (I) 24–*28.9* ± 4.4–32 μm (II) 54–*72.3* ± 8.7–81 μm | 56.6–*64.3* ± 6.4–72.3 × 18–*23.8* ± 2.8–27.4 μm |
| *This study* (ZMA.POR. P.10822) | Norway/130–150 m | (St) 402–*499.5* ± 60.5–540 × 12–*13.7* ± 1.8–16.1 μm | (St) 645–*756* ± 88–1026 × 12.5–*19.3* ± 1.9–21 μm | (I) 23–*27.6* ± 4.1–30 μm (II) 51–*70.1* ± 9.2–78 μm | 52.2–*58.8* ± 7.9–74 × 23.4–*25.9* ± 2.8–30 μm |
| *This study* (ZMA.POR.4977) | Norway/130–150 m | (St) 462–*515.5* ± 54.8–582 × 11.9–*14.2* ± 1.6–16.5 μm | (St) 601.3–*719.5* ± 79.3–1002 × 13.3–*18.2* ± 2.7–22.7 μm | (I) 24–*29* ± 2.6–33 μm (II) 60–*71.5* ± 7.1–84 μm | 48–*55.6* ± 6.2–72 × 24–*25.9* ± 2.4–30 μm |
| ***Melonanchora intermedia* sp. nov.** | | | | | |
| *Pulitzer-Finali (1983)* | Corsica, Mediterranean Sea*/128 m | (St) 380–*490* × 6–11 μm | Same as in ectosome | (I) 19–21 μm (II) 32–49 μm | 37–43 μm |

(Continued)

| Author | Loc./Depth | Ectosomal megascleres | Choanosomal megascleres | Isochelae | Spherancorae |
|---|---|---|---|---|---|
| Reexamination *This study* (MSNG R.N. N IS.4.7) | Corsica, Mediterranean Sea*/128 m | (St) 369–*411.8* ± 14.5–475.3 × 7.2–*9.7* ± 1.5–11 µm | Same as in ectosome | (I) 19–*21.5* ± 0.7–22.7 µm (II) 30.1–*35.2* ± 2.9–38.6 µm (III) 33.2–*39.5* ± 5.1–47.8 µm | 38.9–*44.4* ± 6.7–51.2 × 20–*21.8* ± 1.9–24.2 µm |
| *Díaz, Ramírez-Amaro & Ordines (2021)* | Mallorca Channel, Mediterranean Sea/ 104–138 m | (T) 359–*446*–556 × 5– *8* –11 µm | Same as in ectosome | (I) 14–*18*–21 µm (II) *nm* (III) 29–*42* –47 µm | 36–*40*–46 × 14–*19*–23 µm |
| **Melonanchora insulsa sp. nov.** | | | | | |
| *Schmidt (1880)* | Gulf of Mexico*/ 'deep-sea' | - | – | (I) 23 µm (II) 68 µm | 60 µm |
| Reexamination *This study* (MZS Po165) | Gulf of Mexico*/ 'deep-sea' | (St) 593.6–*656.7* ± 36.2–701 × 16.1–*17.1* ± 1.2–19.5 µm | (S) 813.4–*989* ± 41.2–1121.7 × 19.3–*20.7* ± 1.4–22.5 µm | (I) 27.2–*30.9* ± 3.4–35.8 µm (II) 48.6–*52.3* ± 5.1–68 µm | 52.9–*56.5* ± 4.2–62.1 × 22–*24.3* ± 1.7–26.6 µm |
| **Melonanchora maeli sp. nov.** | | | | | |
| *This study* (ZMA.POR.7269) | Cape Verde*/'deep-sea' | (T) 531.6–*590.9* ± 37.9–627.9 × 9.7–*10.3* ± 0.5–10.6 µm | (S) 637.6–*918.5* ± 75.6–1062.6 × 17.3–*19.2* ± 1.3–21.3 µm | (I) 17.4–*19.8* ± 1.7–23.2 µm (II) 27–*29.3* ± 1.2–31.9 µm (III) 45.4–*49.6* ± 2–53.1 µm | 48.3–*50.2* ± 1.7–53.2 × 17.4–*19.2* ± 1.5–21.3 µm |
| **Hanstoreia globogilva (*Lehnert, Stone & Heimler, 2006a*)** | | | | | |
| Lehnert et al. (2006a) | Aleutian Islands*/ 190 m | (T) 640–680 × 10–12 µm | (Ac) 660–670 × 20–30 µm | (I) 23–25 µm (II) *nm* | (I) 65–93 µm (II) 65–93 µm |
| Reexamination *This study* (NMNH-USNM 1082996) | Aleutian Islands*/ 190 m | (T) 598.9–*675* ± 22.5–724.5 × 9.7–*10.9* ± 2.2–14.5 µm | (Ac) 589.3–*638.3* ± 30–677.3 × 27–*28* ± 1.1–29 µm | (I) 23.1–*25.2* ± 1.1–27 µm (II) 48–*64.4* ± 6.8–67.6 µm | (I) 77.3–*86.9* ± 2.8–91.8 × 27–*30* ± 2.3–33.8 µm (II) *nm* |

**Notes:**
(S) indicates styles; (St) indicates strongyles; (T); indicates tylostyles.
* indicates this is the holotype of the species; *nm* indicates a spicular type that was not mentioned on a description, yet it is assumed was present on the samples.

Holotype (here designated): GNM Porifera 624, Kostergrundet, Sydkoster Island, Sweeden, 100 m depth.

Additional specimens examined:
NHMUK–Icelandic Coll. 1958.1.1.633, Iceland, North Atlantic Ocean (63.55, −11.41666), 1936; NHMUK Norman Coll. 1898.5.7.38, Norway, 1893; NHMUK, 83.12.13.70.89; MZLU L935/3858, Koster, Säcken, Sweeden, Baltic Sea (59.00971, 11.11471), 1934, (*Alander, 1935*; *1942*); ZMA.POR.P.10796, Northwest of Tromsø, Norway, Arctic Ocean (72.60138, 24.95), *R/V Willem Barents* expedition (1880–84), 256 m depth, 1881 (*Vosmaer, 1885*); ZMA.POR.P.10825, Marsteinsboen, Norway, North East Atlantic (60.12583, 4.98944), 130–150 m depth, on stone, 1982; ZMA.POR.P.10822, Marsteinsboen, Norway, North East Atlantic (60.12583, 4.98944), 130–150 m depth, on stone, 1982; ZMA.POR. P.10824, Marsteinsboen, Norway, North East Atlantic (60.12583, 4.98944), 130–150 m depth, on stone, 1982; ZMA.POR.4977, Marsteinsboen, Norway, North East Atlantic

(60.12583, 4.98944, 130–150 m depth, on stone, 1982; ZMA.POR.P.10823, off Saengsbokt, Bergen, Norway, North East Atlantic (60.36666, 4.81666), 350–600 m depth, 1982; ZMA. POR.4976, off Saengsbokt, Bergen, Norway, North East Atlantic (60.36666, 4.81666), 350–600 m depth, 1982.

Unregistered material:
NR0509_82b, Flemish Cap, Tail Grand Bank, North Atlantic Ocean, 1,127 m depth (NEREIDA Coll.); NR0610_30a, Flemish Cap, Tail Grand Bank, North Atlantic Ocean, 613 m depth (NEREIDA Coll.).

Description:
Massive-globular sponge, with an easily detachable paper-like thin ectosome bearing abundant fistular processes (typical of the genus). The choanosome is orange-cream in colour and the ectosome results whitish, yet translucent, in alcohol.

Skeleton:
Spicule arrangement as in the other species of the genus (viz. *M. elliptica*), with its main distinguishing feature being the presence of strongyles as choanosomal megascleres.

Spicule complement:
Tylostrongyles, strongyles, two categories of isochelae, and spherancorae (Figs. 7A–7F)

Ectosomal tylostrongyles (Fig. 7B): As in other *Melonanchora*, they are slightly flexuous, with a more or less central swelling. The tips can be strongyloid or slightly tylote often vaguely unequal.
Size range: 483–*542.6* ± 38.3–600 µm × 10.6–*12.9* ± 3.2–19.3 µm

Choanosomal strongyles (Fig. 7A): Entirely smooth, with asymmetrical ends (one clearly rounded and the other blunt but somewhat narrower. More or less curved throughout its entire length.
Size range: 627.9–*802.3* ± 42.2–924.5 µm × 11.6–18.3 ± 1.5–24.4 µm

Isochelae I (Fig. 7E, c'): Anchorate, with a straight shaft, gently bending to its ends, with three-spatulated alae.
Size range: 21.2–*26.5* ± 3.8–28.9 µm

Isochelae I (Fig. 7D, b'): Similar to isochelae I, but smaller in size.
Size range: 48.6–*68.6* ± 8.1–72.9 µm

Spherancorae (Fig. 7C, a'): With a prolate-oval shape, and dentate fimbriae on its internal face, which might be free or fused at various degrees. The junction points of each couple of opposite alae can be observed in most spicules, with the resulting fused shaft being slightly asymmetrical.
Size range: 48.3–*67.5* ± 6.8–78.62 × 18.9–*22.3* ± 1.6–25.2 µm

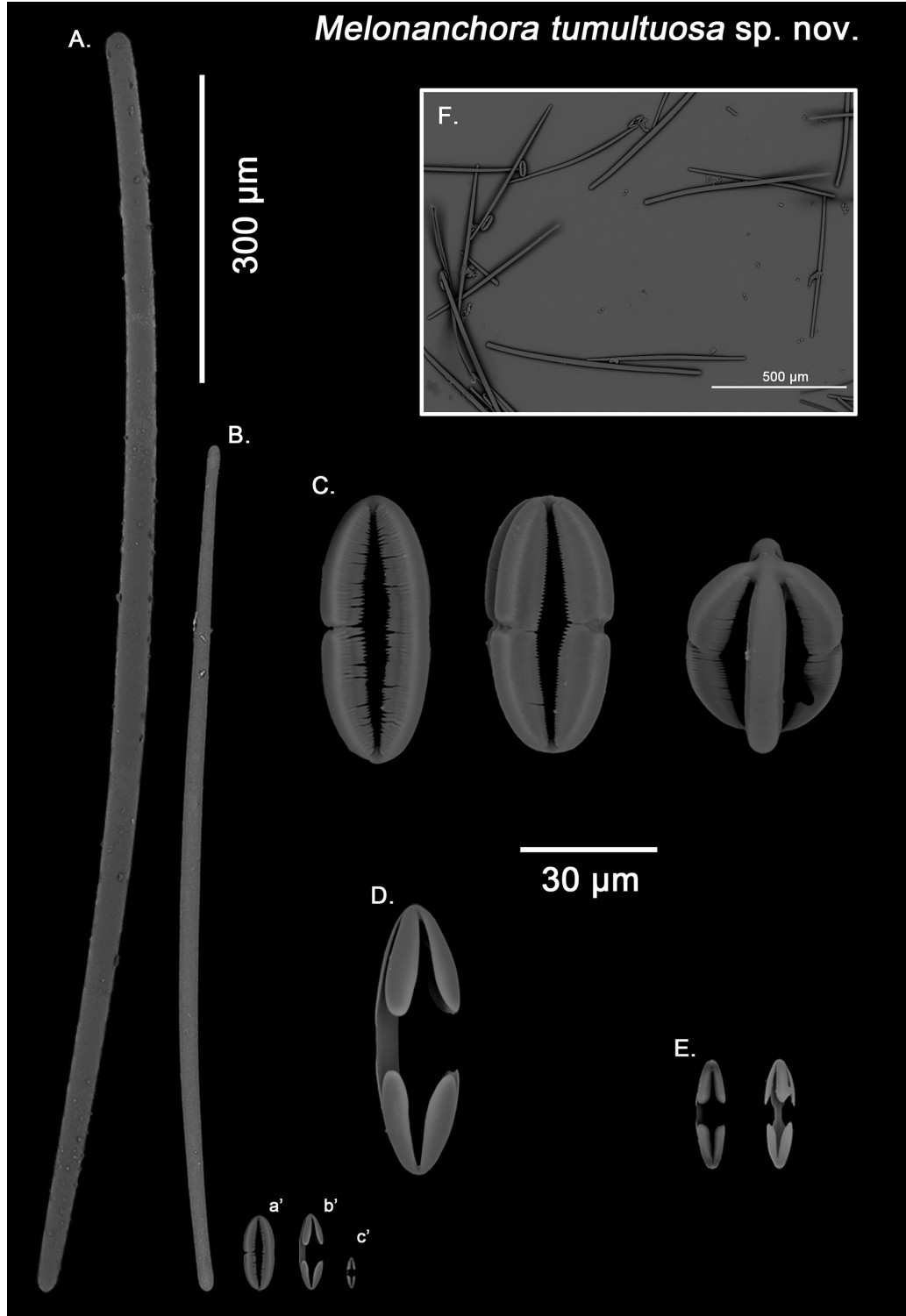

**Figure 7** *Melonanchora tumultuosa* **spicule plate.** Spicular set for *Melonanchora tumultuosa* sp. nov. (sample GNM Por 624, holotype). (A) Choanosomal strongyle; (B) Ectosomal tylostrongyle; (C) Spherancorae; (D) Large chelae category (Chelae II); (E) small chelae category (Chelae I), (F) General view of *M. tumultuosa* sp. nov. spicules by SEM imaging. (a') Spherancora (b') Chelae II and (c') Chelae I relative sizes when compared with that of the megascleres. Scale bars for (A), (B), (a'), (b'), (c') 300 μm; (C), (D), (E) 30 μm and (F) 500 μm.

Geographic distribution and type locality:
The species presents an amphi-Atlantic distribution (Fig. 4), being sympatric with
*M. elliptica*. Its type locality is the Sydkoster Island, Sweden, yet, known records for the
species also include Iceland (NHMUK–1958.1.1.633) the Davis Strait (*Baker et al., 2018*)
and Norwegian coasts (*Vosmaer, 1885*; this paper).

Etymology:
From the latin *tumultuosus*, meaning full of commotion. It refers to the confusion that
samples of this species have caused between *M. elliptica* and *M. emphysema* during the past
century.

Remarks:
Specimens of *M. tumultuosa* sp. nov. had been considered by several authors to be
*M. emphysema* because of their possession of both ectosomal and choanosomal strongyles
(*Baker et al., 2018*). Close re-examination of the *M. emphysema* type revealed only one
type of megascleres, which is present in both ectosome and choanosome (Fig. 5A), whereas
in *M. tumultuosa* sp. nov., two different types of strongyles characterise either the
ectosome (Fig. 7B) or the choanosome (Fig. 7A).

   Additionally, it had been suggested that those *Melonanchora* with two strongyle
categories could in fact be *M. elliptica* individuals with styles modified into strongyles
(*Baker et al., 2018*). In this regard, sponge spicules might vary in shape due to
environmental conditions (*Bell, Barnes & Turner, 2002*) and/or silica abundance (*Uriz
et al., 2003*) even to the point not expressing one or more spicule types (*Maldonado &
Uriz, 1996*; *Maldonado et al., 1999*). However, *M. elliptica* and *M. tumultuosa* sp. nov.
co-occur in their areas of distribution, even at local scales (*Baker et al., 2018*), weakening
such an idea. Finally, *M. tumultuosa* sp. nov., spherancorae shape is mostly prolate
(Fig. 7C), commonly with asymmetrical shafts and rounded ends, whereas they are clearly
spheroidal in *M. elliptica*, with slightly pointed ends (Fig. 3C), which is translated in an
overall slender spherancorae for *M. tumultuosa* sp. nov, compared to *M. elliptica* (average
width 29.7 *vs*. 22.3 µm respectively; Tables 1 & 3).

### *Melonanchora intermedia* sp. nov.
(Fig. 8)

Synonymy:
*Melonanchora emphysema*; *Pulitzer-Finali, 1983*: 561; *Díaz, Ramírez-Amaro & Ordines,
2021*: 42–43, fig. 16.
Not *Melonanchora emphysema* (*Schmidt, 1875*: 118).

Material examined.
Holotype (here designated): MSNG Vis4.7–off Calvi, Corsica (42.53333, 8.6), depth 128 m,
detrital, dredge, 18 July 1975. R.N. N IS.4.7 (*Pulitzer-Finali, 1983*).

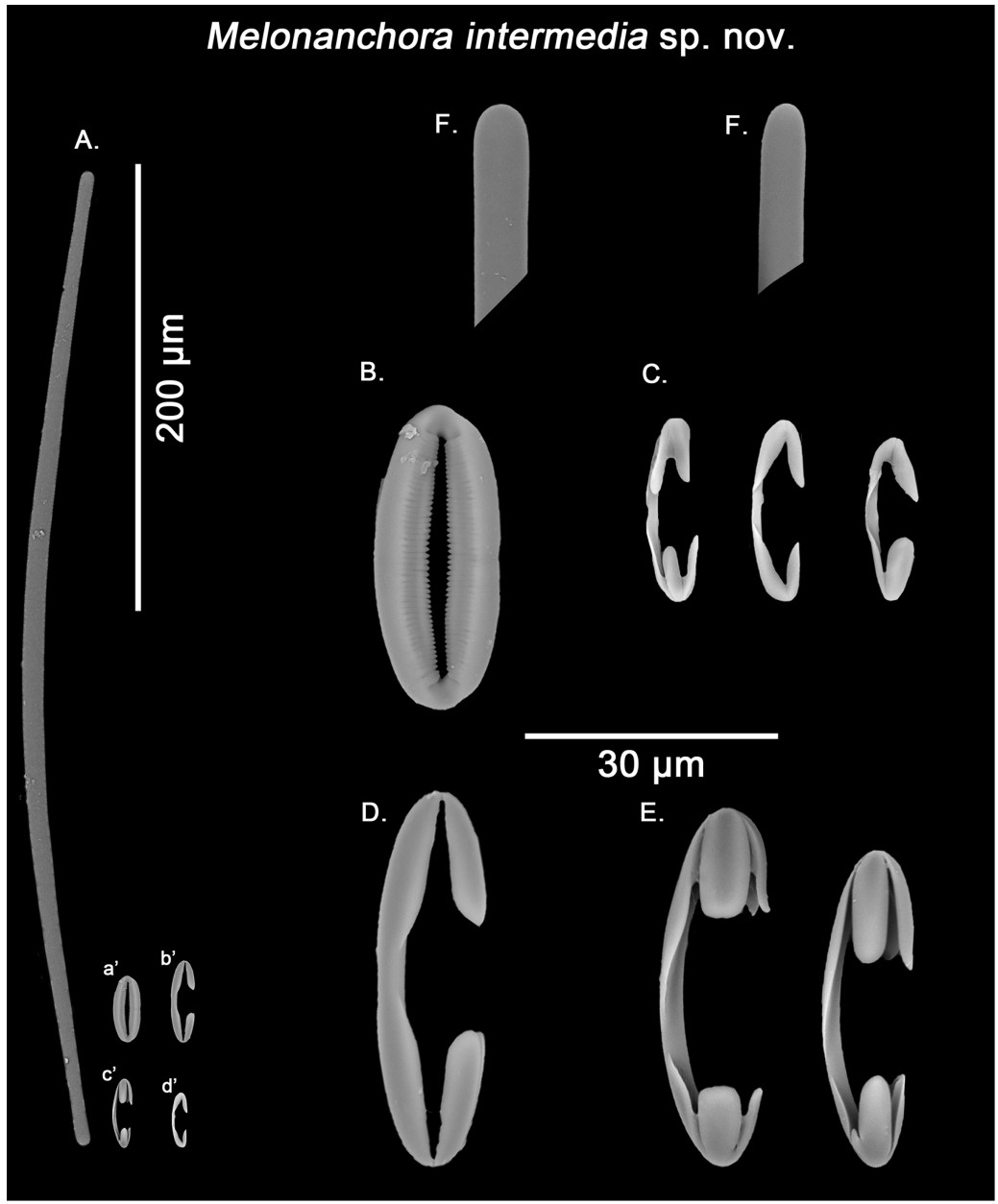

**Figure 8** *Melonanchora intermedia* **spicule plate.** Spicular set for *Melonanchora intermedia* sp. nov. (sample MSNG Vis4.7, holotype). (A) Ectosomal and chonasomoal tylostrongyle; (B) Spherancorae; (C) small chelae category (Chelae I); (D) Large chelae category (Chelae II); (E) Anisochelae; (F) Detail of the tylostrongyle's ends. (a') Spherancora (b') Chelae II (c') Anisochelae and (d') Chelae I relative sizes when compared with that of the megascleres. Scale bars for (A), (a'), (b'), (c'), (d') 200 µm; (B), (C), (D), (E), (F) 20 µm.

Description:

Small subglobular individual attached to rocky debris. It possesses a paper-like ectosome with the warty-like papillae typical of the genus, yet with just a few papillae.

Skeleton:
Ill-defined paucispiculate tracts in the choanosomal area, and a clear crisscross pattern can be observed in the ectosome. Microscleres are abundantly scattered throughout the choanosome.

Spicule complement:
Tylostrongyles, three categories of chelae and spherancorae (Figs. 8A–8F).

Ectosomal and choanosomal tylostrongyles (Fig. 8A): from more or less straight to entirely bent on its length. The show a wider central zone, narrowing asymmetrically toward differently marked tylotoid ends (Fig. 8F), giving the spicule a variable shape between strongyles to tylostrongyles.
Size range: 369.6–*411.8* ± 14.5–475.3 μm × 7.2–*9.7* ± 1.5–11 μm

Isochelae I (Fig. 8C, d'): anchorate, with a gently curved shaft and irregularly spatulated rounded alae, often with a malformed tooth in one or both of the extremes.
Size range: 19–*21.5* ± 0.7–22.7 μm

Isochelae II (Fig. 8D, b'): With an almost straight shaft and three alae, presenting a prominent fusion between the lateral alae and the shaft.
Size range: 30.1–*35.2* ± 2.9–38.6 μm

Isochelae III (Fig. 8E, c'): With a long, gently curved shaft and slightly asymmetrical ends, *e.g.*, the alae of one extreme are ca. 1.5 longer that those of the opposite extreme (anisochelae appearance). Alae are usually flat and with a straight end (occasionally with a bifid appearance), occupying ca. ¼ of the spicule size.
Size range: 33.2–*39.5* ± 5.1–47.8 μm

Spherancorae (Fig. 8B, a'): with an elongated shape, and fimbriae on its internal face, which can be free or fused to varying degrees. Spherancorae with incompletely fused alae are present.
Size range: 38.9–*44.4* ± 6.7–51.2 × 20–*21.8* ± 1.9–24.2 μm

Geographic distribution and type locality:
The species seems so far to be endemic to the Mediterranean Sea (Fig. 4), having only been recorded from its type locality off Calvi, on the Corsica island (*Pulitzer-Finali, 1983*) and, more recently, from the Mallorca Channel (*Díaz, Ramírez-Amaro & Ordines, 2021*). Regarding its ecology, while records are still scarce it appears to occur at rhodolith beds and rocky environments close to the limit of the continental shelf, between 104 to 134 m depth.

Etymology:
From the Latin *intermedia* ("in between"). The name refers to its unique possession of a third intermediate category of isochelae, contrary to almost all other *Melonanchora* species, which only possess two.

Remarks:
The closest species to *M. intermedia* sp. nov. would be *M. emphysema*, a typical deep-sea species also recorded from the Mediterranean Sea. Both species share the presence of tylostrongyles as their only megascleres, yet their microscleres present clear divergences, with isochelae being smaller in size in *M. intermedia* sp. nov. compared to *M. emphysema* (avgerage length (I) 21.5 and (II) 35.2 *vs.* (I) 24.7 and (II) 60.2 μm, respectively; Tables 2 & 3), as well as the presence of a third category of chelae with flat, slightly asymmetrical ends in *M. intermedia* sp. nov. In this sense, in their description of 'Melonanchora emphysema' *Díaz, Ramírez-Amaro & Ordines (2021)* only mentions two chelae categories with no apparent aberrant morphologies, which could cast doubts about its placement between *M. emphysema* or *M. intermedia* sp. nov. Nevertheless, all spicular categories mentioned in *Díaz, Ramírez-Amaro & Ordines (2021)* fall within the size range of *M. intermedia* sp. nov. (Table 3), and its biggest isochelae category possess flat ends, which is one of the defining characteristics of *M. intermedia* sp. nov. Regarding the fact that only two chelae categories could be identified in his specimen, it is possible that isochelae II and III might have been confused in optical microscopy as, in fact, Fig. 16 of that same publication depicts a isochelae with rounded alae which matches in size (ca. 35 μm) the isochelae II category of the holotype. It is also interesting to note that the smallest isochelae category in the holotype of *M. intermedia* sp. nov. usually showed alae with aberrant morphologies (Fig. 8C), a feature that was not described for the Mallorca specimen. As so, this might point out that the presence and/or abundance of certain of chelae types within this species might be subjected to a certain degree of intraspecific variation. Finally, the Mallorca specimen shares with *M. intermedia* sp. nov. a subglobular appearance, as well as depth range and habitat (100–140 m depth) which further supports its inclusion as *M. intermedia* sp. nov. as opposed to *M. emphysema*, which is appears to be an encrusting sponge mostly limited to the deep-sea and other cold-water environments.

*Melonanchora insulsa* sp. nov.
(Fig. 1E; 9)

Synonymy:
*Melonanchora elliptica Schmidt, 1880*: 85, pl. IX fig. 8.
Not *Melonanchora elliptica Carter, 1874*: 212.

Material examined.
Holotype (here designated): MZS Po165, Gulf of Mexico, *USCSS Blake* expedition (1878–79) in the Gulf of Mexico, (24, −86), deep-sea dredging, 1879.

Description:
A small (less than 1 cm$^2$), thin fragment of choanosome, and some scrapped pieces of ectosome (Fig. 1E). Although we cannot report on the sponge's original shape, Schmidt 1 (880) described the sample as a crust growing on an euplectellid glass sponge from the genus *Regadrella*.

Skeleton:
The ectosomal skeleton consists of tangential strongyles with a criss-cross arrangement, whereas the choanosomal skeleton is formed by ill-defined style-made tracts. Microscleres are widespread throughout the choanosome without a clear discernible pattern.

Spicule complement:
Styles, strongyles, two categories of chelae, spherancorae (Figs. 9A–9F).

Ectosomal strongyles (Fig. 9B): slightly flexuous, with more or less unequal ends.
Size range: 593.6–*656.7* ± 36.2–701 × 16.1–*17.1* ± 1.2–19.5 µm

Choanosomal styles (Fig. 9A): entirely smooth, mostly straight, with acerate points (Fig. 9F), sometimes slightly curved towards its distal end.
Size range: 813.4–*989* ± 41.2–1121.7 × 19.3–*20.7* ± 1.4–22.5 µm

Isochelae I (Fig. 9E): Smaller in size, and with a more prominent fusion between the lateral alae and the shaft.
Size range: 27.2–*30.9* ± 3.4–35.8 µm

Isochelae II (Fig. 9D): With a gently curved shaft, and spatulated alae.
Size range: 48.6–*52.3* ± 5.1–68 µm

Spherancorae (Fig. 9C): with an elliptical slightly asymmetrical shape, and teeth-like fimbriae on its internal face, which might be free or fused to different extent. Ridges of the spherancorae are unequally, gently bent, giving its ellipsoid shape a slightly asymmetrical appearance.
Size range: 52.9–*56.5* ± 4.2–62.1 × 22–*24.3* ± 1.7–26.6 µm

Geographic distribution and type locality:
The species is so far only known from the Gulf of Mexico (East of the Campache Escarpment, 24.0°N 86.0°W), and was collected from deep waters (Fig. 4).

Etymology:
From the latin *in-* ("not") + *salsus* ("salted"), meaning insipid, tasteless. The name refers to the original description of the specimen made by *Schmidt (1880)*, who regarded the sample as boring or "*uninteressanten*".

Remarks:
*Schmidt (1880)* unambiguously stated that this individual from the Gulf of Mexico belonged to *M. elliptica*. However, the two types of chelae in *M. elliptica*'s have a straight shaft with free alae pointing outwards (Figs. 3D–3E), whereas in *M. insulsa* sp. nov. chelae show a slightly bent shaft and its alae are more parallel to the later (Figs. 9D–9E). Apart from their morphological differences, the smaller isochelae category appears to be bigger in *M. insulsa* sp. nov. (average length 30.9 µm; Table 3) when compared with those from *M. elliptica* (average length 26.6 µm; Table 1). Moreover, *M. elliptica*'s spherancorae are regularly oval (Fig. 3C), whereas *M. insulsa*' spherancorae are irregular,

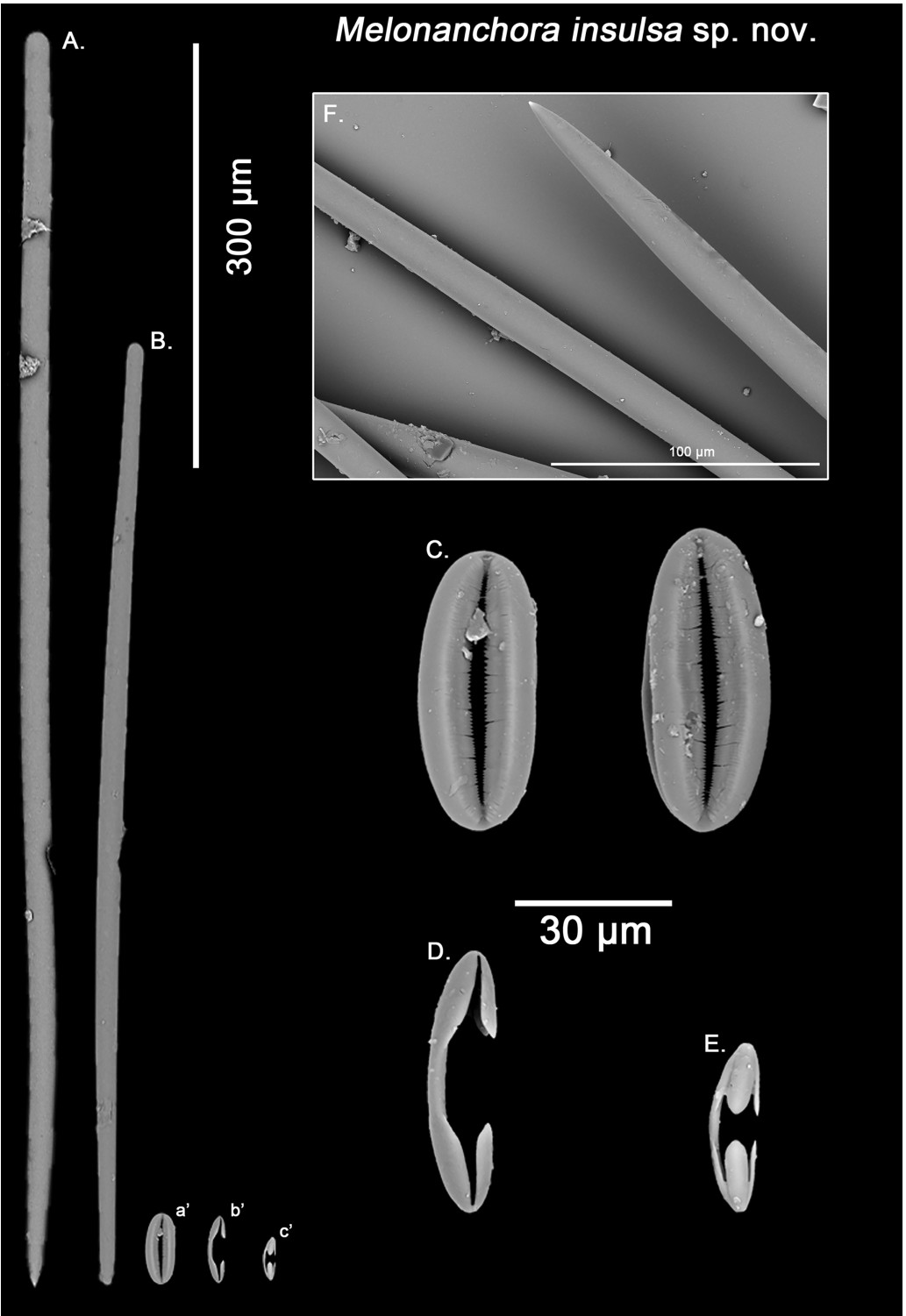

**Figure 9 *Melonanchora insulsa* spicule plate.** Spicular set for *Melonanchora insulsa* sp. nov. (sample MZS Po165, holotype). (A) Choanosomal style; (B) Ectosomal tylostrongyle; (C) Spherancorae; (D) Large chelae category (Chelae II); (E) small chelae category (Chelae I); (F) Detail of the styles' acerate end as seen in SEM imaging. (a') Spherancora (b') Chelae II and (c') Chelae I relative sizes when compared with that of the megascleres. Scale bars for (A), (B), (a'), (b'), (c') 300 µm; (C), (D), (E) 30 µm and (F) 100 µm.

somewhat asymmetrical ellipsoids (Fig. 9C), supporting the distinction of *M. insulsa* sp. nov. as a different species from *M. elliptica*.

<p align="center">*Melonanchora maeli* sp. nov.<br>(Fig. 1G; 10)</p>

Synonymy:
*Melonanchora emphysema*; *van Soest, 1993*: 210, Tab. 2.
Not *Melonanchora emphysema* (*Schmidt, 1875*: 118).

Material examined.
Holotype (here designated): ZMA.POR.7269, Ponta Tremorosa, Ilha de Santiago, Cape Verde, (14.8833, −23.5333), 1986; ZMA.POR.P. 10826, Ponta Tremorosa, Ilha de Santiago, Cape Verde, (14.8833, −23.5333), 1986 (microscopic slide).

Description:
A small sub-globular sponge, covered with abundant, proportionally big, bulbous fistules which arise from a paper-thin like ectosome (Fig. 1G). The ectosome is only attached here and there to the cavernous choanosome, making the former easily detachable. The choanosome is beige-orange and the ectosome is somewhat whitish, yet translucid.

Skeleton:
The ectosomal skeleton consists of tangential tylotes with a more or less developed criss-cross arrangement, whereas the choanosomal skeleton is formed by ill-defined style-made tracts. Microscleres are widespread thorough the choanosome without a clear discernible pattern.

Spicule complement:
Styles, tylotes, three categories of chelae and spherancorae (Figs. 10A–10H). The sample was contaminated with tetractinellid spicules from an unidentified specimen stored altogether with the holotype.

Ectosomal tylostrongyles (Fig. 10B): slightly flexuous, with clearly marked tyles at both ends. Very regular in size.
Size range: 531.3–590.9 ± 37.9–627.9 × 9.7–10.3 ± 0.5–10.6 μm

Choanosomal styles (Fig. 10A): entirely smooth and mostly straight to slightly bent, always with acerate endings. The heads vary between those of true styles to true tylostyles (Fig. 10G), albeit the later are rare.
Size range: 637.6–918.5 ± 75.6–1062.6 × 17.3–19.2 ± 1.3–21.3 μm

Isochelae I (Fig. 10F; d'): Small anchorate chelae, with a straight, short shaft, long fimbriae and spatulated alae.
Size range: 17.4–19.8 ± 1.7–23.2 μm

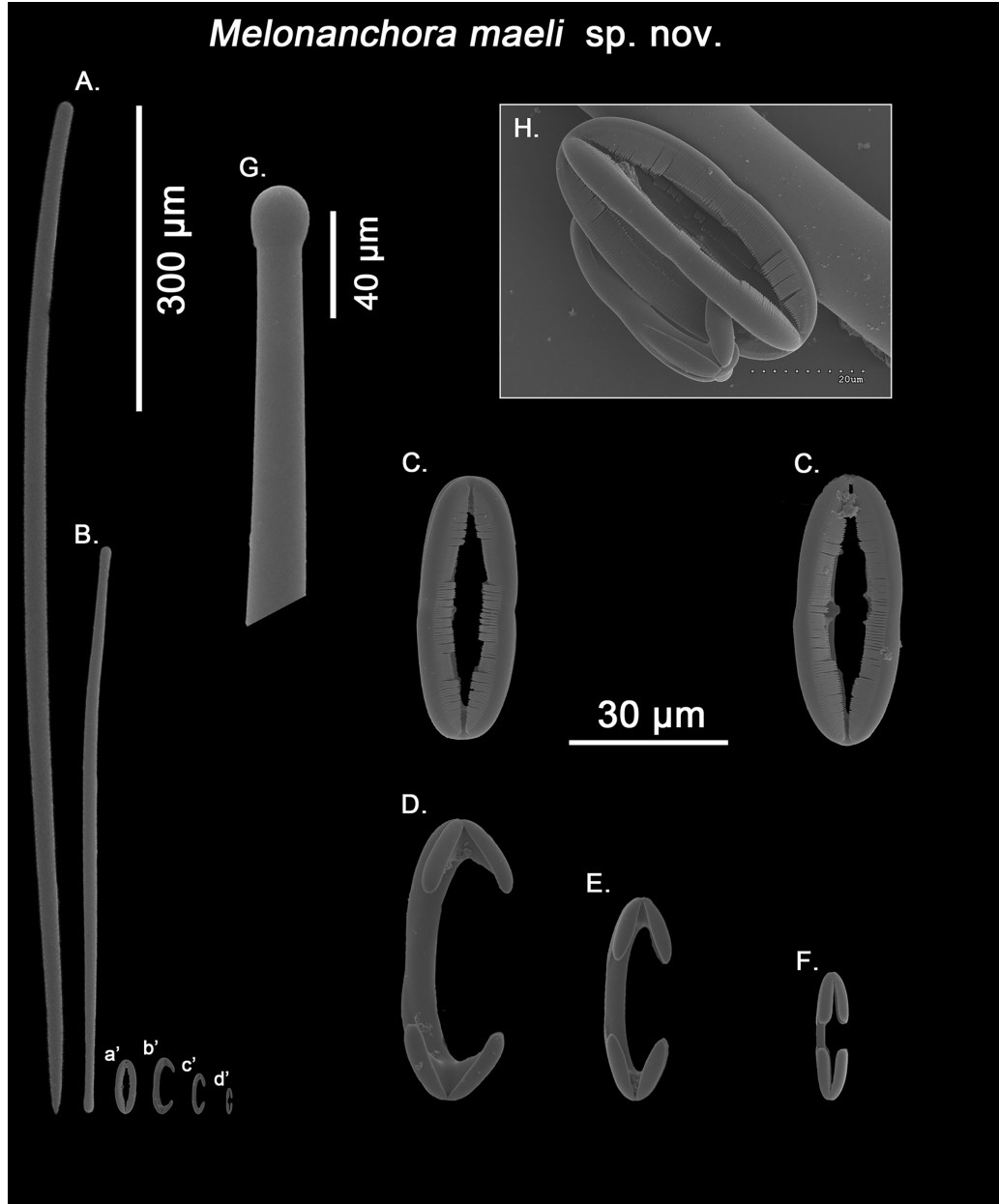

**Figure 10 *Melonanchora maeli* spicule plate.** Spicular set for *Melonanchora maeli* sp. nov. (sample ZMA.POR.7269, holotype). (A) Choanosomal style; (B) Ectosomal tylostrongyle; (C) Spherancorae; (D) Large chelae category (Chelae III); (E) Intermediate chelae category (Chelae II); (F) Small chelae category (Chelae I); (G) Head of a style modified into a tylostyle; (H) Detail of a spherancora lateral view. (a') Spherancorae (b') Chelae III (c') Chelae II and (d') Chelae I relative sizes when compared with that of the megascleres. Scale bars for (A), (B), (a'), (b'), (c'), (d') 300 μm; (C), (D), (E), (F) 30 μm; (G) 400 μm and (H) 20 μm.

Isochelae II (Fig. 10E; c'): The least abundant of all three chelae categories, with a slightly bent shaft, in intermediate size between isochelae I and III, with short, slender alae. Only 29 spicules could be measured.

Size range: 27–*29.3* ± 1.2–31.9 μm

Isochelae III (Fig. 10D; b'): The biggest of the three isochelae categories, it is strikingly similar to isochelae II, with a long, slightly bent shaft and reduced slim alae. Yet, the alae are more reduced in regards to the general size of the spicule, and they are widely opened in respect to each other, contrary to isochelae II, where the separation between alae isn't obvious.
Size range: 45.4–*49.6* ± 2–53.1 μm

Spherancorae (Figs. 10C, 10H; a'): with an elongated oval shape, almost straight with just a subtle curvature near the tips, and teeth-like fimbriae on its internal face. It usually shows a slightly asymmetrical appearance.
Size range: 48.3–*50.2* ± 1.7–53.2 × 17.4–*19.2* ± 1.5–21.3 μm

Geographic distribution and type locality:
This is the southernmost species of *Melonanchora* known to date, and, the only species of the genus to occur in Cape Verde archipelago (14° 52′ 59.88″N 23° 31′ 59.88″W) (Fig. 4).

Etymology:
The species is dedicated to *Mael*, the Elder God of the Seas in the world of Malaz, co-created by Steven Erikson and Ian C. Esslemont, in recognition of the vast and unique universe of their novels.

Remarks:
Originally identified as *M. emphysema* (van Soest, 1993), the specimen appears to be new to science. While its spicule complement would place it close to *M. elliptica* or *M. insulsa* sp. nov. due to the possession of styles as choanosomal megascleres, the presence of three chelae categories easily tells it apart from those. Additionally, the shape of the chelae is very different to that of the abovementioned species, with considerably reduced alae in two of the chelae categories (Figs. 10D and 10E), a feature which isn't shared by any other *Melonanchora* species. Furthermore, its spherancorae are almost straight (Fig. 10C), whereas in most other *Melonanchora* species a clear oval morphology can be observed.

## Genus *Hanstoreia* gen. nov.

Diagnosis:
Massive-globular growth form, with paper-like, easily detachable thin ectosome, bearing multiple fistular processes. Ectosomal skeleton going from no apparent apparent organization to an ill-defined crisscross of smooth strongyles to tylotes with somewhat asymmetrical ends, whereas the choanosome is composed of ill-defined acanthostyles tracts. Microscleres include typically two categories of anchorate isochelae, rarely three, with at least one in the form of acanthose, incomplete 'spherancorae'.

Type species:
*Melonanchora globogilva* Lehnert, Stone & Heimler, 2006a: 9–13 (here designated).

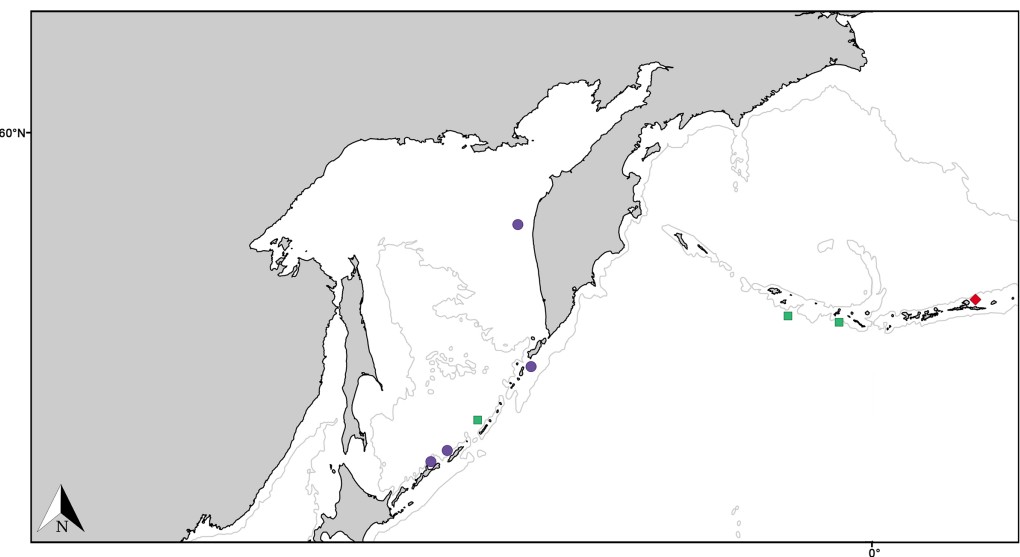

**Figure 11 Distribution map for North Pacific species previously in *Melonanchora*.** Distribution map for *Hanstoreia globogilva* (red diamond), *Myxilla (B.) kobjakovae* (green square) and *Arythmata tetradentifera* (purple circle). Projected view (UTM Zone 31N (WGS84)) with geographic (WGS84) coordinates indicated for reference. A grey line represents the 1,000 m depth isobaths. Geographic and bathymetric data used was obtained from http://www.naturalearthdata.com.

Etymology:

The genus is dedicated to a much esteemed and dearly missed Nordic colleague, Hans Tore Rapp (University of Bergen), in recognition of his exceptional contributions on taxonomy and ecology of deep-sea sponges of the boreal and Arctic regions.

Remarks:

*Hanstoreia globogilva* was recently described from the Pacific Ocean (Fig. 11), and tentatively assigned to the genus *Melonanchora*, yet it presented some unique spicule types absent from their Atlantic counterparts (*Lehnert, Stone & Heimler, 2006a*). In this sense, the species clearly resembled *M. elliptica*, the type species of *Melonachora*, yet it possessed acanthostyles (Fig. 12A) as choanosomal megascleres and particular isochelae with dentate fimbria (Fig. 12C) along the internal face of its alae and shaft, which were reminiscent of spherancorae, the main diagnostic feature for *Melonanchora*.

The placement of this species within *Melonanchora* was initially based on its external morphology (Fig. 1F) and, under the consideration that other *Melonanchora* species (viz. *M. tetradedritifera Koltun, 1970* and *M. kobjakovae Koltun, 1958*) had been previously described with incomplete 'spherancorae' (*Koltun, 1958, 1970*). However, SEM observation of Koltun's species (this study, Figs. 13–14) proved that those species did not bear true spherancorae but more or less complete cleistochelae, and therefore both *M. tetradedritifera* and *M. kobjakovae* need to be reassigned to other genera (See below).

While the dissimilarities between *H. globogilva* and *Melonanchora* are quite clear (smooth *vs.* acanthose choanosomal megascleres, complete *vs.* incomplete 'spherancorae'), they also share several traits (mainly two categories of smooth isochelae, ectosomal tylostrongyles to strongyles, a thin translucent paper-like ectosome and a more or less subspherical external morphology) thus, arguments both in favour and against erecting a new genus for *H. globogilva* could be made. Regarding *H. globogilva*'s external appearance, within Poecilosclerida there are other unrelated genera apart from *Melonanchora* (viz. *Cornulum*, family Acarnidae; *Coelosphaera*, family Coelosphaeraeidae) which might present a subglobular appearance and possess a warty, paper-like ectosome. Thus, external appearance alone does not represent a reliable character for genus assignation.

Regarding its spicular complement and skeletal arrangement, *H. globogilva* is indubitably closer to *Melonanchora* than to any other genera within Myxillidae, but it still presents major differences with the former. In this sense, *H. globogilva* possesses true acanthostyles, which are lacking from any other *Melonanchora* representative so far. Furthermore, all known *Melonanchora* possess an ectosomal arrangement of tangential tylostrongyles forming a dense, well-defined crisscross pattern (Fig. 2C), whereas in the choanosome megascleres are mostly arranged in spicule tracts, with some free spicules in between. On the contrary, in the case of *H. globogilva* tylostrongyles in the ectosome are mostly arranged in a confused manner, whereas their choanosomal tracts are ill-defined and with abundant free spicules in between. Nevertheless, the main difference between *H. globogilva* and *Melonanchora* would be that of its supposed incomplete 'spherancorae'. In this sense, *H. globogilva* possesses unique, acanthose square-shaped chelae, which might be reminiscent of spherancorae while still in formation. Nevertheless, it has already been proven that unique microscleres, including chelae derivatives, might have evolved independently by phylogenetically distant species. In this sense, a similar case to that of *H. globogilva* and *Melonanchora* would be that of the proposed synonymy of *Abyssocladia Lévi, 1964* with *Phelloderma Ridley & Dendy, 1886* by *van Soest & Hajdu (2002)*. *Abyssocladia* was known from just three ill-known species while *Phelloderma* was monotypic, but both genera appeared to share the possession of a unique, apparently identical chelae type in the form of 'abyssochelae'. Nevertheless, and as noted by the authors, both species greatly differed in all other aspects, including general shape, skeletal architecture and the rest of its spicular complement. The discovery of additional species of *Abyssocladia* casted additional doubts about the genus status, which was then revived and reassigned to the family Cladorizhidae based on its similar skeletal arrangement, presence of sigmancistras and shared carnivorous habit (*Vacelet, 2006*). Finally, the use of molecular markers demonstrated that *Abyssocladia* and *Phelloderma* were not closely related (*Vargas et al., 2013*), and thus that, despite their striking similarity, their unique chelae had developed independently. Lastly, it is also worth noticing that if included in *Melonanchora*, *H. globogilva* would be the sole representative of the genus in the Pacific, whereas all other species occur in the North Atlantic.

While *Melonanchora* and *H. globogilva* could be arguably closer to each other than to other Myxillidae, based on previous precedents we have decided to erect a new genus, *Hanstoreia* gen. nov. to allocate *H. globogilva*, rather than including it in *Melonanchora*.

*Hanstoreia globogilva* (*Lehnert, Stone & Heimler, 2006a*)
(Figs. 1F; 12)

Synonymy:
*Melonanchora globogilva Lehnert, Stone & Heimler, 2006a*: 9–13, fig. 4a–4f, fig. 5a–5d;
*Stone, Lehnert & Reiswig, 2011*: 88, Apendix IV. 168–169.
*Melonanchora globoblanca Lehnert, Stone & Heimler, 2006a*: 12 (misspelling of the former).

Material examined.
Holotype: NMNH-USNM 1082996, north of Amlia Island, Aleutian Islands (58.46902, −173.59802), 190 m depth, 2006.

Description:
Sub-spherical shape, with an easily detachable paper-like thin ectosome bearing abundant bulbous fistules (Fig. 1F). The choanosome is light-yellow and the ectosome is somewhat translucent-whitish, in life.

Skeleton:
The ectosomal skeleton consists on a loose crisscross of spicules arranged perpendicularly to the surface here and there, yet for most of it no clear arrangement can be discerned. The choanosome consists of ill-arranged tracts of tylotes and acanthostyles, without a clear discernible orientation, and with the tylotes being restricted to the upper areas of the choanosome. Microscleres are abundant and concentrate towards the choanosomal tracts.

Spicule complement:
Tylotes, acanthostyles, and three chelae categories, one of them in the form of incomplete 'spherancorae' (Figs. 12A–12F).

Ectosomal tylotes (Fig. 12B): Unevenly flexuous, with a central thickening, unequally thinning towards both ends, which sow variable tyles with variable swellings.
Size range: 598.9–*675* ± 22.5–724.5 × 9.7–*10.9* ± 2.2–14.5 µm

Choanosomal acanthostyles (Fig. 12A): Slightly curved along its length, with an acerate point. Spines are short and stout, moderately abundant along the entire shaft but the tip.
Size range: 589.3–*638.3* ± 30–677.3 × 27–*28* ± 1.1–29 µm

Isochelae I (Fig. 12E): with a straight shaft, well-developed fimbriae and spatulated alae, the lateral ones largely fused with the shaft.
Size range: 23.1–*26.2* ± 1.1–27 µm

Isochelae II (Fig. 12D): Almost identical to isochelae I, but bigger in size.
Size range: 48.3–*64. 4* ± 6.8–67.6 µm

Spherancorae (Fig. 12C): Uncompleted, with free teeth, resembling chelae. As in all other *Melonanchora*, dentate fimbriae cover its internal face.
Size range: 77.3–*86.9* ± 2.8–91.8 × 27–*30* ± 2.3–33.8 µm

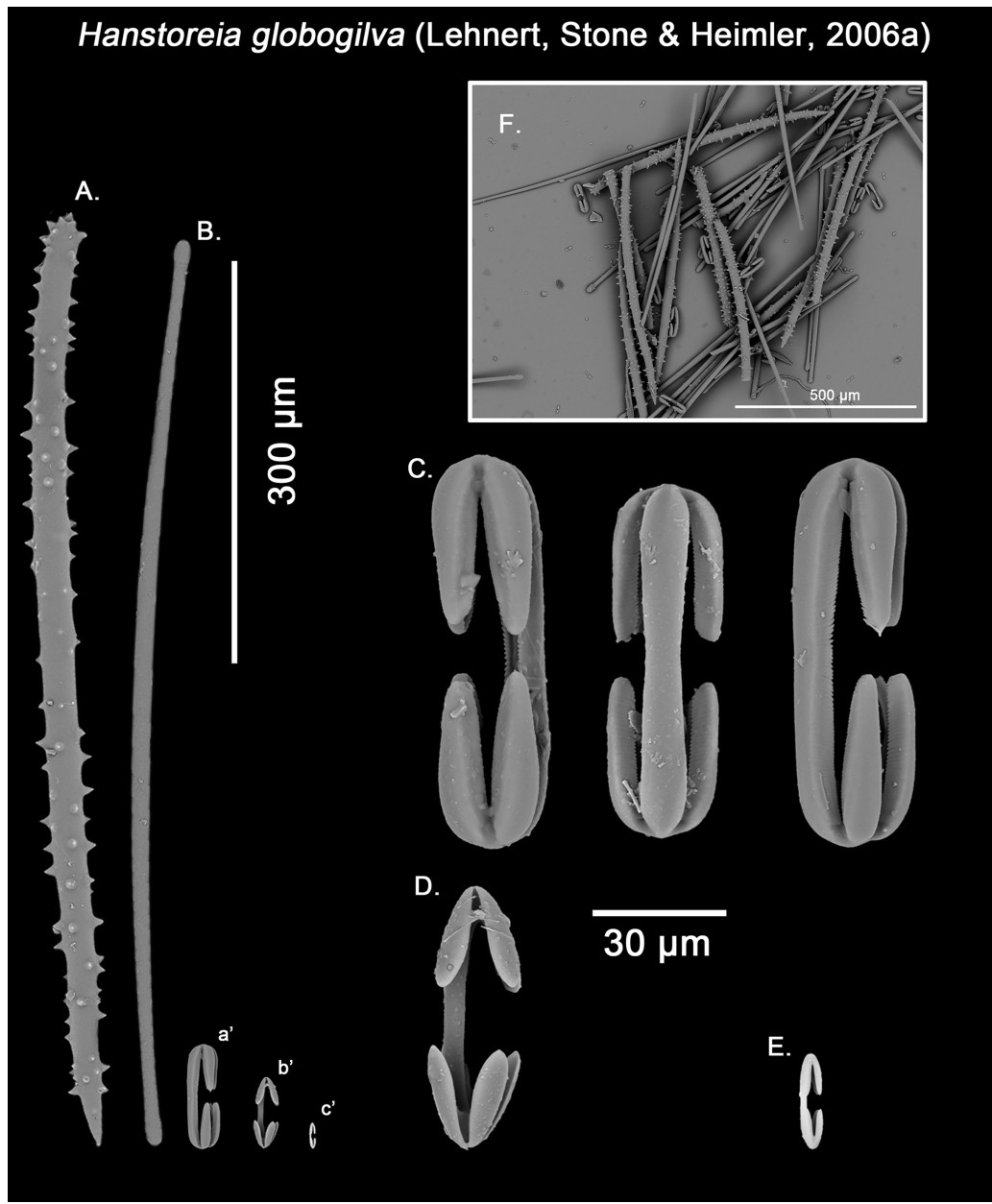

**Figure 12 *Hanstoreia globogilva* spicule plate.** Spicular set for *Hanstoreia globogilva* (sample NMNH-USNM 1082996, holotype). (A) Choanosomal acanthostyle; (B) Ectosomal tylostrongyle; (C) Spherancorae; (D) Large chelae category (Chelae II); (E) small chelae category (Chelae I); (F) General view of *H. globogilva*'s spicules by SEM imaging. (a') Spherancora (b') Chelae II and (c') Chelae I relative sizes when compared with that of the megascleres. Scale bars for (A), (B), (a'), (b'), (c') 300 μm; (C), (D), (E) 30 μm and (F) 500 μm.

Geographic distribution:
The species appears to be rare, as it has only been seldomly recorded from deep bottoms around the Aleutian Archipelago (*Lehnert, Stone & Heimler, 2006a*; *Stone, Lehnert & Reiswig, 2011*) (Fig. 11).

Remarks:

The original description mentions a second category of spherancorae-isochelae with outer dented margins which could not be found again upon re-examination of the type material. As they are in the same size-range as the incomplete 'spherancorae', they are here regarded as likely to constitute aberrant modifications or developmental stages of *H. globogilva*'s unique chelae. Additionally, the re-examination of the type material made it clear the existence of a second, larger, isochelae category almost identical to its smallest one but much less abundant, which might explain its absence from the species' original description.

Genus *Myxilla* Schmidt, 1862
Subgenus (*Burtonanchora*) Laubenfels, 1936

Type species:
*Myxilla (Burtonanchora) crucifera* Wilson, 1925 (by original designation).

Diagnosis:
*Myxilla* with smooth choanosomal styles. Chelae are three-teethed, with occasional polydentate modifications (amended from van Soest, 2002).

*Myxilla* (*Burtonanchora*) *kobjakovae* (Koltun, 1958)
(Fig. 13)

Synonymy:
*Melonanchora kobjakovae* Koltun, 1958: 58, fig. 13; Koltun, 1959: 122, fig. 75; pl. XVII, fig. 4; pl. XVIII, fig. 2; Javnov, 2012 (*partim*): 65–66.

Material examined:
Syntype (here designated): NHMUK 1963.7.29.23, Southern Kuril Islands, Pacific coast, *R/V Toporok* Kuril-Sakhalin expedition (1946–49) (Stns 127, 128), Deep-sea dredging, 1949. Exchanged with V. M. Koltun in July 1963.

Description:
The sponge is tubular, digitate or funnel shaped, with a long stem. Its surface is smooth, with the oscules being located on the top of the finger-like processes in the digitate forms. Colour bright orange in life, and from ochre to dark-brown, in alcohol.

Skeleton:
Choanosomal skeleton consisting of a dense isodyctial reticulation of multispicular tracts embedded in spongin fibres without echinating spicules. Ectosomal skeleton formed by a tangential layer of more or less disarranged spicules.

Spicule complement:
Styles, strongyles, and two categories of chelae (Figs. 13A–13E).

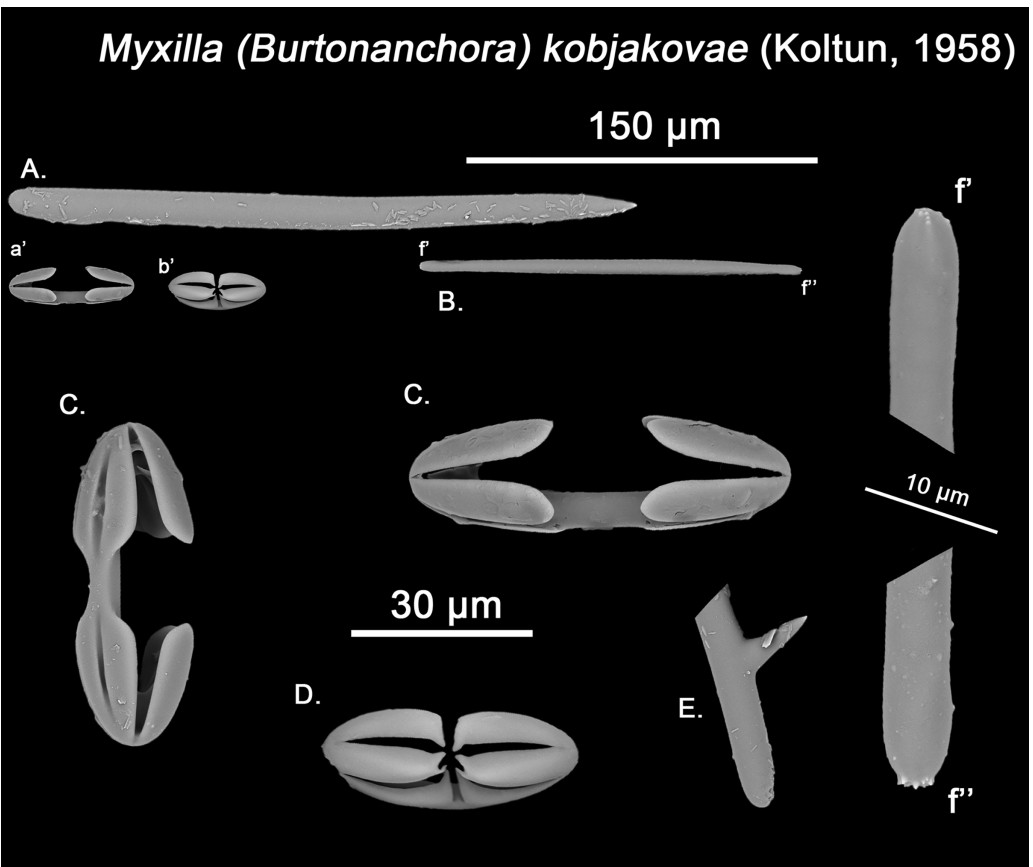

**Figure 13 Myxilla kobjakovae spicule set** Spicular set for *Myxilla (B.) kobjakovae* (sample NHMUK 1963.7.29.23, holotype). (A) Choanosomal style; (B) Ectosomal strongyle; (C) Large chelae category (Chelae I); (D) Small chelae category (Chelae II); (E) Style's aberrant end; (f') close up view of the strongyles microspinned end; (f'') close up view of the strongyles' microspinned other end. (a') Chelae I (b') Chelae II relative sizes when compared with that of the megascleres. Scale bars for (A), (B), (a'), (b') 150 μm; (C), (D), (E) 30 μm and (f'), (f'') 10 μm.

Ectosomal strongyles (Fig. 13B): Straight, short and stout, with a subtle swelling at each end (Fig. 13f', f''), finished in a ring of weak spines, typical of *Myxilla*. They can also be found scattered through the choanosome.
Size range: 140.3–*190.3*–323.8 ± 12.2 × 7.1–*9.8*–12.5 ± 2.1 μm

Choanosomal styles (Fig. 13B): slightly curved along its length, with and acerate distal end and a proximal end sometimes vaguely inflated.
Size range: 327.5–*397.5*–567.3 ± 23.2 × 17.8–*20.3*–22.6 ± 1.9 μm

Isochelae I (Fig. 13D): Unusual small anchorate isochelae with three prominent alae ending in a double hook-like termination. The alae of both ends almost contact each other, somewhat resembling a cleistochelae. Fimbriae are well developed, and present and inner hook on its lower part which point towards the interior of the chelae.
Size range: 29.2–*33.3*–35.7 ± 2.8 μm

Isochelae II (Fig. 13C): Anchorated, three-teethed chelae, with spatulated alae. It has clear, well developed fimbriae, which expand from the shaft.
Size range: 60.1–*79.7*–87.6 ± 7.8 μm

Geographic distribution:
So far, the species has only been recorded from the Okhotsk Sea, at the Kuril, Iturup and Urup islands (*Koltun, 1958*, *1959*; *Javnov, 2012*; *Guzii et al., 2018*) and the Kamchatka peninsula (*Calkina, 1969*) at depths ranging from 28 to 231 m (Fig. 11).

Remarks:
*Myxilla* (*B.*) *kobjakovae* was initially assigned to *Melonanchora* based on the presence of smooth choanosomal megascleres and spherancorae (*Koltun, 1958*). Yet, after re-examining the holotype, we verified that those supposed spherancorae were in fact cleistochelae derivatives (Fig. 13D). Additionally, *M. kobjakovae* clearly deviates from *Melanonchora* species in growth form, lack of a paper-like ectosome, and type of megascleres. Besides *Melonanchora*, just two other Myxillidae genera possess smooth megascleres: *Myxilla* (*Burtonanchora*) *Laubenfels, 1936* and *Stelodoryx Topsent, 1904*. Both genera resemble each other in most aspects (*Lehnert & Stone, 2015*), yet *Stelodoryx* is defined as possessing polydentate anchorate isochelae whereas *Myxilla* (*B.*) has exclusively three- teethed anchorate isochelae (*van Soest, 2002*). However, *Myxilla* (*B.*) *asigmata Topsent, 1901* has been observed to possess chelae with 3–5 alae (*Ríos & Cristobo, 2007*), implying that the definition of *Myxilla* should be modified to include the eventual possession of polydentate chelae. On the other hand, as a result of the inclusion of some other genera as synonyms of *Stelodoryx* by *van Soest (2002)*, some of the current species of *Stelodoryx* possess three-teethed chelae (*viz. Stelodoryx lissostyla* (*Koltun, 1959*). As to, whether *Stelodoryx* and *Myxilla* are synonymous or two different genera is unclear and in need of a taxonomic revision.

The presence of polydentate chelae, while not specific enough, is still used as the main classifying feature to distinguish *Myxilla* and *Stelodoryx* (*Bertolino et al., 2007*; *Lehnert & Stone, 2015*). Thus, the new species is here referred to *Myxilla* (*Burtonanchora*) due to the possession of three-teethed anchorate chelae, yet it differs from most other *Myxilla* (*B.*) in the absence of sigmas, possession of two chelae categories, one of them in the form of cleistochelae, and its stalked growth form. Further reclassification of the species should not be ruled out in light of a broader Myxillidae review.

Finally, the species description in the Russian Fauna of the East seas (*Javnov, 2012*) depicts varying morphologies for *M. kobjakovae*. While polymorphism is common in sponges, the huge variations depicted in the Russian individuals, which range from the typical digitate-branching orange sponge, to conical-shaped or tubular-rimmed, cream coloured individuals (*Javnov, 2012*) suggest they may represent a different related species.

### Genus *Arhythmata* gen. nov.

Type species:
*Melonanchora tetradedritifera Koltun, 1970* (here designated).

Diagnosis:
Lamellate sponge, apparently resulting from coalescent digitations, with the surface slightly uneven. Ectosome thin, coriaceous, easy to detach, with subectosomal cavities. Oscula are large and unevenly spread. Choanosome crossed by numerous canals. The ectosomal skeleton is a tangential layer of strongyles perpendicular to the choanosomal spicule tracts. The choanosomal skeleton consists of a loose isodyctial reticulation of multispicular style tracts embedded in spongin. The spicule complement consists of smooth choanosomal styles, ectosomal tylotes with spiny heads and three categories of polydentate chelae, among which, at least one is asymmetrically modified. So far, monotypic genus restricted to the deep-sea areas around the Okhotsk Sea.

Etymology:
From the Latin *arhythmatus*, meaning "inharmonious" or "of unequal measure", referring to the asymmetry of the alae of *A. tetradedritifera*'s peculiar chelae.

Remarks:
*Arhythmata tetradedritifera* was originally described as *Melonanchora tetradedritifera* based on the possession of smooth choanosomal styles, two categories of chelae, and spherancorae (*Koltun, 1970*). However, Koltun misidentified unique, modified chelae as spherancorae (See "The Origin of Spherancorae"), and described styles and tylostrongyles that highly differed in shape from those of other *Melonanchora* species. This spicule combination draws the species closer to *Myxilla (Burtonanchora)* and *Stelodoryx* as they are the only Myxillidae genera with smooth styles. However, in contrast to *M. (B.) kobjakovae*, *A. tetradedritifera* possesses polydentate (4–5) chelae, which will place the species closer to *Stelodoryx* than to *Myxilla*. However, while *Myxilla (Burtonanchora)* (13 accepted species; *van Soest et al., 2021*) represents a narrowed, well-defined, portion of *Myxilla* (91 accepted species; *van Soest et al., 2021*), *Stelodoryx* (18 accepted species; *van Soest et al., 2021*), represents an amalgam of spicule types on a rather small genus (*Lehnert & Stone, 2015*). Indeed, the actual concept of *Stelodoryx* is only distinguished from *Myxilla* by the presence of polydentate chelae, yet little attention has been paid to the other spicule complements (*Lévi, 1993*). Megascleres in *Stelodoryx* include both smooth (viz. *Stelodoryx flabellata Koltun, 1959*) or spiny (viz. *Stelodoryx mucosa Lehnert & Stone, 2015*) ectosomal tylotes or tornotes, or even styles (viz. *Stelodoryx siphofuscus Lehnert & Stone, 2015*); with choanosomal acanthostyles (viz. *S. mucosa*), smooth styles (viz. *S. siphofuscus* or *S. mucosa*), microspined styles (viz. *Stelodoryx lissostyla* (*Koltun, 1959*)), oxeas (viz. *Stelodoryx oxeata Lehnert, Stone & Heimler, 2006a*, *2006b*) or even strongyles (viz. *S. flabellata*). Additionally, chelae may be three-teethed (viz. *S. lissostyla*) or polydentate, with teeth varying from four to seven, having from one (viz. *S. flabellata*) to three (viz. *S. oxeata*) chelae categories, with occasional accompanying sigmas (viz. *S. oxeata* or *S. mucosa*). Thus *Stelodoryx*, with just 18 species, harbours a spicule variability that might equal those of all four subgenera of *Myxilla* together (*van Soest, 2002*). With a combination of strongyles with microspined head and smooth styles, the closest relative to *A. tetradedritifera* within *Stelodoryx* would be *Stelodoryx jamesorri*

*Lehnert & Stone, 2020* which has already been signalled as of difficult allocation within the genus *Stelodoryx* (*Lehnert & Stone, 2020*). While both species share several common traits (stout choanosomal smooth styles, ectosomal tylotes to strongyles with microspined heads and the possession of two categories of peculiar polydentate chelae), both species differ in the possession of third, unique chelae category for *A. tetradedritifera* and in their skeletal organization, being plumoreticulate in *Stelodoryx jamesorri*, as opposed to the isodyctial reticulation observed in *A. tetradedritifera*. Finally, *Stelodoryx pluridentata* (*Lundbeck, 1905*) and *Stelodoryx strongyloxeata Lehnert & Stone, 2020*, would also be arguably close to *A. tetradedritifera*, but they possess ectosomal styles instead of strongyles (*Lévi, 1993*; *Lehnert & Stone, 2020*), with additional sigmas in the former (*Lévi, 1993*) and choanosomal strongyleoxeas in the later (*Lehnert & Stone, 2020*).

Consequently, a new genus, *Arhythmata* gen. nov., is here erected to properly accommodate *Melanonchora tetradedritifera*, with a diagnosis based on the combination of ectosomal microspined strongyles, smooth choanosomal styles in an isodyctial arrangement, and three polydentate chelae categories and, from which at least one is modified into an asymmetrical chelae, a rare feature within Poecilosclerida, which has been considered of taxonomic value for other genera (*e.g., Echinostylnos* spp.; *Lévi, 1993*), and which are here termed *retortochelae* (Fig. 14C) and defined as "*asymmetrical stout chelae in which alae are not facing their direct opposite, but the space in-between opposing alae*". Interestingly enough, retortochelae appear to be very rare within Porifera, with *Echinostylnos Topsent, 1927* being the only other genera with asymmetrically twisted chelae, albeit not all its accepted species possess such (*Carvalho et al., 2016*). On the other hand, their stout, somewhat clesitocheliferous morphology is also relatively unusual within chelae, just being common in two other genera: *Abyssocladia* (known as abyssochelae) and *Phelloderma*. Despite their rarity, molecular analyses have shown that said chelae have been independently acquired (*Vargas et al., 2013*; *Göcke, Hajdu & Janussen, 2016*), thus being safe to assume that this is also the case for *Arhythmata* gen. nov. Finally, while currently the genus remains monotypic, this might change in the future upon a proper re-examination of the genus *Stylodoryx*, which is much in need of revision.

*Arhythmata tetradedritifera* (*Koltun, 1970*)
(Figs. 1D, 14)

Synonymy:
*Melonanchora tetradedritifera Koltun, 1970*: 209, fig. 22.

Material examined.
NMNH-USNM 148959, AB120069, South of Amlia Island, Central Aleutian Islands, Pacific coast, (51.8392, −173.906), 337 m depth, July 2012; NMNH-USNM 1478958, AB120046, South of Kanaga Island, Central Aleutian Islands, Pacific coast, (51.5587, 177.622), 358 m depth, July 2012.

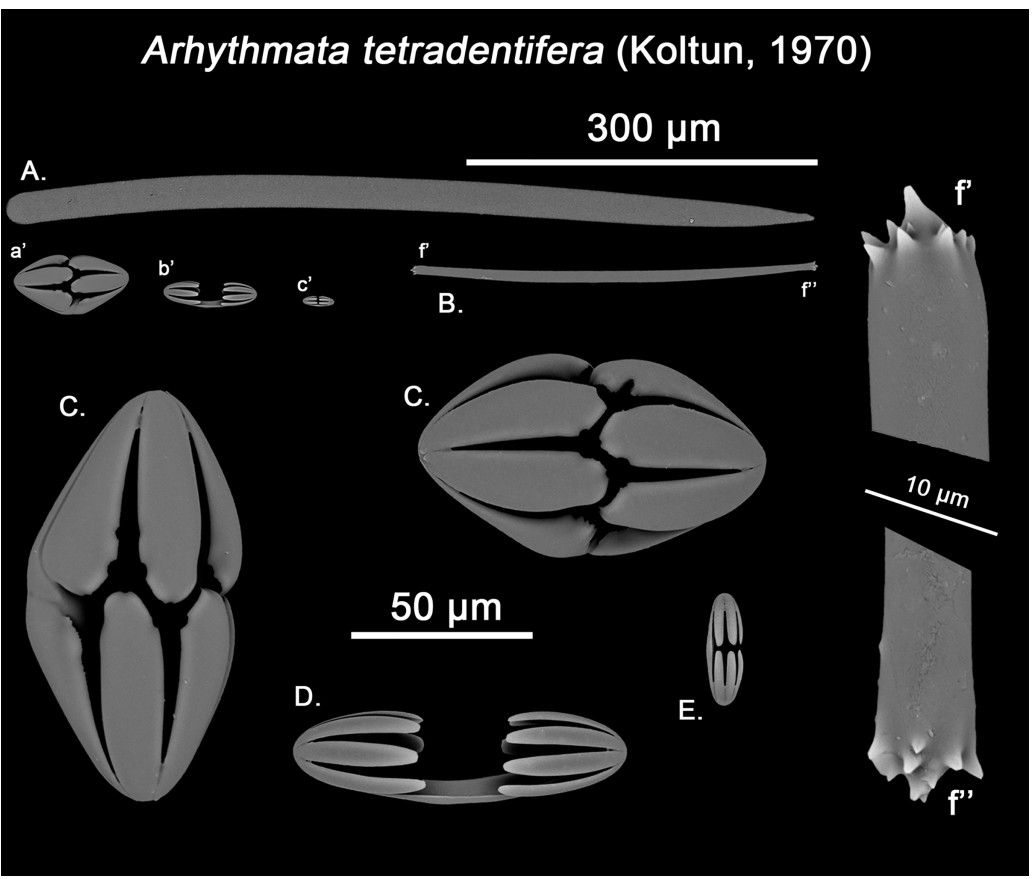

**Figure 14 _Arhythmata tetradentifera_ spicule plate.** Spicular set for _Arythmata tetradentifera_ (sample NMNH-USNM 148959). (A) Choanosomal style; (B) Ectosomal strongyle; (C) and (C') Retortochelae; (D) Large chelae category (Chelae II); (E) Style's aberrant end; (f') close up view of the strongyles microspinned end; (f'') close up view of the strongyles' microspinned other end. (a') Retortochelae (b') Chelae II and (c') Chelae I relative sizes when compared with that of the megascleres. Scale bars for (A), (B), (a'), (b'), (c') 300 μm; (C), (D), (E) 50 μm and (f'), (f'') 10 μm.

Description:
As described in the genus definition (Fig. 1D). All the examined samples contained sand grains through the choanosome. Additionally, the colour when dry is dark brown, close to kobicha (kelp like) or taupe (brown-greyish), whereas the ectosome is whitish with wheat-like shadings.

Skeleton:
The ectosomal skeleton consists of a somewhat confused tangential layer of strongyles perpendicular to the choanosomal spicule tracts, which consists of a loose isodyctial reticulation of multispicular style tracts embedded in spongin.

Spicule complement:
Styles, strongyles, three categories of chelae (Figs. 14A–14D).

Ectosomal strongyles (Fig. 14B): Short, straight, with both ends slightly spinose and slight inflated somewhat unequally (Fig. 14f', f''); a distal thorn is present, which gives them the appearance of tornote-like strongyles.
Size range: 270.5–*307.8*–357.4 ± 24.3 × 9.6–*10.3*–14.5 ± 1 µm

Choanosomal styles (Fig. 14A): Entirely smooth, slightly curved along its length, almost doubling in width the tylostrongyles.
Size range: 521–*608*–685 ± 54.3 × 24.1–*29.3*–33.8 ± 2.3 µm

Isochelae I (Fig. 14E): Small ancorate pentadentate, with a short shaft.
Size range: 48.3–*60.4*–67.7 ± 7.3 µm

Isochelae II (Fig. 14D): ancorate pentadentate isochelae, with a comparatively large, almost straight shaft.
Size range: 67.7–*70.6*–87.3 ± 3.4 µm

Retortochelae (Fig. 14C): Asymmetrical, almost ovoid, ancorate isochelae with a curved, somewhat twisted shaft and four five, long teeth. The upper and lower teeth are not facing each other but slightly displaced, in such a way that each tooth occupies the space between two opposite teeth and vice versa. This makes the chelae asymmetrical, with the alae looking as if they have been sculpted with notches and tips to accommodate the opposing alae.
Size range: 77.3–*88.6*–106 ± 2 × 48.3–*49.1*–53.1 ± 2 µm

Geographic distribution:
Currently, the species has only been located at the deep-sea waters (338 to 3,335 m depth) of the Okhotsk Sea, mostly around the Simushir Islands (*Koltun, 1970*; *Downey, Fuchs & Janussen, 2018*) and the Aleutian Islands (Fig. 11).

Remarks:
Although the holotype of this species could not be examined, the studied material fits well with Koltun's original description, in terms of spicule types and sizes (*Koltun, 1970*). However, the species has been observed to possess two different chelae categories, mainly distinguished by its size and shaft lengths, which were not described by Koltun, while the spherancorae mentioned in the original description are, in fact, modified chelae with a twisted shaft, long teeth and an ovoid contour (retortochelae; Fig. 14C).

*Arhythmata tetradedritifera* represents a new addition to the already diverse Myxillidae fauna of the Okhotsk deep-sea and nearby areas. During the past years, several new species from the area have been included in Myxillidae (*Lehnert, Stone & Heimler, 2006a*, *2006b*; *Lehnert & Stone, 2015*), which might partially respond to a high abundance of endemic benthic fauna in the area (*Downey, Fuchs & Janussen, 2018*). Although the genus remains monotypic for the time being, further exploration in the deep bottoms of the Okhotsk Sea and nearby areas might result in the discovery of additional species.

## DISCUSSION

### Diversity and biogeography of the genus *Melonanchora*

In contrast to most sponge genera, *Melonanchora* shows a quite narrow distribution, restricted to the circumpolar Arctic and some North Atlantic areas. Additionally, only one species, *M. elliptica* could be considered common across its distribution area (*Fristedt, 1887*; *Lundbeck, 1905*; *van Soest & De Voogd, 2015*; *Baker et al., 2018*). Despite initial misidentification of fossil spherancorae (*Hinde & Holmes, 1892*), there are no known fossil records for the genus, thus making discussion about its origin and radiation, tentative.

Contrary to biogeographic distributions of other sponge genera, which suggest they may have a Tethyan or Gondawanan origin (*e.g., Acarnus*, *van Soest, Hooper & Hiemstra, 1991*; *Rhabderemia*, *van Soest & Hooper, 1993*; *Hajdu & Desqueyroux-Faúndez, 2008*; *Hamigera*, *Santín et al., 2020*), the current distribution of *Melonanchora* might be better explained by trans-Arctic exchanges. The opening of the Bering Strait during the late Pliocene (ca. 5.3 Ma; *Vermeij, 1991*), allowed a massive interchange of species among northern areas of the Atlantic and the Pacific (*Vermeij, 1991*), which is supported by both the fossil record (*Reid, 1990*) and molecular studies (*Dodson et al., 2007*; *Coyer et al., 2011*). This exchange did not just occur among vagile fauna (*Dodson et al., 2007*), but also among benthic species (*Reid, 1990*), including sponges (*Ereskovsky, 1995*). Benthic species are known to have crossed the strait, in the several opening and closing events of the strait during the glacial and interglacial periods (*Coyer et al., 2011*). Additionally, during these glacial and interglacial periods, species expanded or contracted their distribution areas as a result of climate changes and their associated biotic and abiotic factors, which provided new suitable habitats (*Jansson & Dynesius, 2002*). The common ancestor for both *Melonanchora* (Atlantic) and *Torentendalia* gen. nov. (Pacific), might have expanded from Pacific to Atlantic waters during one of the several events that opened the Bering Strait, with the aforementioned genera resulting from the isolation of its Pacific and Atlantic populations. Once in the Atlantic, it could have expanded further south towards the tropical regions during the glacial periods (*Ereskovsky, 1995*). Thus, *M. maeli* sp. nov. and *M. insulsa* sp. nov., the only representatives of the genus close to the equator, might be a legacy of this latitudinal migration, being now confined to "deep-sea refugia" due to previous climatic changes (*Ereskovsky, 1995*; *Convey et al., 2009*). Finally, the Mediterranean *M. intermedia* sp. nov. might represent a recent speciation from *M. emphysema*, which might have entered the Mediterranean after the Messinian Salinity Crisis, as hypothesized for other Mediterranean sponges (*Boury-Esnault, Pansini & Uriz, 1992*; *Xavier & Van Soest, 2012*). However, the lack of fossil records in their current distribution area (*Ereskovsky, 1995*) and the lack of phylogenetic data, paired with the scarcity of material of most *Melonanchora* species, makes it difficult to properly assess the vicariant events that led to its diversification, leaving the field open for future research efforts.

## The origin of spherancorae

The order Poecilosclerida *Topsent, 1928*, build around the exclusive presence of chelae is, with over 2.500 formally described species (*van Soest et al., 2021*) possibly the most diverse group within Porifera (*Hooper & Van Soest, 2002*). The high taxon diversity parallels that of its chelae, with basic chelae morphotypes (palmate, anchorate, and arcuate) described for the first time by *Levinsen (1893)* and *Lundbeck (1905, 1910)*, and several modifications of the formers (*Hajdu, van Soest & Hooper, 1994*; *Hooper & Van Soest, 2002*).

In his initial description of *Melonanchora*, *Carter (1874)* assumed that the two chelae categories present his specimen where in fact early developmental stages of the unique, "melon-shaped" chelae, which characterized the genus or even, the last developmental stage of anchorate chelae (*Vosmaer, 1885*). While this view was soon refuted, and the "melon-shaped" chelae was recognized as a separate chelae type (*Schmidt, 1880*), it was not until 1885 that they were given a specific designation, "*mel*", based on their unique shape (*Vosmaer, 1885*). However, the name would remain unsettled for the following years, with several authors following Vosmaer's proposal as *melonanchoras* (*Fristedt, 1887*; *Levinsen, 1893*; *Arnesen, 1903*), while others followed Topsent's proposed designation (*Topsent, 1892*) of *sphearancisters* (*Thiele, 1903*; *Topsent, 1904*). Topsent's proposal however, was based on his perception that each shaft of the chelae resembled a diancistra (*Topsent, 1892*). However, diancistras are sigmoid derivatives (*Hajdu, van Soest & Hooper, 1994*) whereas spherancorae are true chelae derivatives (*Levinsen, 1893*). Nevertheless, the term "*melonanchora*" was identical to that of the genus, which could lead to confusion. As so, Lundbeck settled the dispute in 1905, when he designated these unique chelae as spherancorae, highlighting its chelae nature and unique oval morphology (*Lundbeck, 1905*).

Regarding the spherancora's unique morphology, the common presence of developmental stages in several individuals has given a proper view of their chelae nature (*Levinsen, 1893*) as well of their developmental stages. Spherancorae start as slim ancorate chelae, with a thin shaft and three teeth (Fig. 15.1), of the same width. Later, those three teeth expand, until they coalesce (Fig. 15a), forming four indistinguishable shafts, all being at approximately right angles in respect to each other, and giving the spherancorae its characteristic oval shape (Fig. 15.2). While not usually visible as they occur on the internal shaft's view, the junction points of the alae usually develop into a swelling in adult spherancorae (Fig. 15c). Right after the arcs are formed, the spherancorae begin the development of its internal "teeth-brims" (Fig. 15.3), as in other teethed chelae, (*e.g., Guitarra solorzanoi*; *Cristobo, 1998*). The internal dentate fimbriae are regularly arranged along the internal surface of the *Melonanchora*'s shaft (Fig. 15.4; 15.5; 15.5'), yet the teeth are not fused to the shafts, but are free and protrude from a small ridge formed at side of the shafts (Fig. 15c). The length and a degree of fusion vary between individuals of the same species, ranging from the most common free teeth forms (Fig. 15b), to partially joined teeth, or even almost coalescent teeth. This intraspecific variability

regarding the fusion degree of the alae might partially reflect silica availability at the time the spicules were formed (*Uriz et al., 2003*), as it has been reported for other sponge taxa (e.g., *Bavestrello, Bonito & Sará, 1993*; *Cárdenas & Rapp, 2013*).

While the spherancora's morphology seems to be rather conservative between *Melonanchora* species, *H. globogilva* possesses unique acanthose chelae, which would resemble incomplete 'spherancorae' (Fig. 12C). These chelae present non-coalescenting alae and internal teeth-brims, which might loosely resemble those in placochelae (*Cristobo, 1998*), yet this is likely to be anecdotal, and of little to no taxonomical significance. Nevertheless, the architecture of this third chelae category could be consistent with that of the developmental stages of true spherancorae (Fig. 15), as its teeth-brims are not restricted to the alae, but are present all along the shaft's internal surface, as true spherancorae. As so, *H. globogilva*'s unique chelae might point towards a common ancestor between both *Melonanchora* and *Toretendalia* gen. nov., and represent, in fact, ancestral 'spherancorae' (*Lehnert, Stone & Heimler, 2006a*), further supporting its chelae ancestry.

Confusion between spherancorae and other spicular types is highly unlikely, yet there are a few spicular types that could, or have been, confused with spherancorae. Placochelae and derivatives (Fig. 16C) are a complex group of microscleres, synapomorphic for the family Guitarridae (*Uriz & Carballo, 2001*; *Hajdu & Lerner, 2002*), which share with spherancorae the possession of teeth-brims along the shafts and alae (*Hajdu, van Soest & Hooper, 1994*). While the possible affinity of Guitarridae with Myxillidae was eventually proposed (*van Soest, 1988*), this was poorly supported, among others, by the likely palmate origin of placochelae (*Hajdu, van Soest & Hooper, 1994*), which are absent in Myxillidae. The development of teeth-brims among chelae, while not a common trait, should be regarded a homoplastic character acquired independently by several taxa. Apart from placochelae, both cleistochelae (*viz. M. (B.) kobjakovae*) and clavidiscs (*Hinde & Holmes, 1892*; *Ivanik, 2003*) have been interpreted at some point as spherancorae due to their ovoidal morphology. Fossil *Merlia* species (viz. *Merlia morlandi* (*Hinde & Holmes, 1892*); *Merlia* sp. *Ivanik, 2003*; *Lukowiak, Pisera & Stefanska, 2019*) have been confused with *Melonanchora* due to the similarity between clavidiscs (Fig. 16D) and spherancorae (Fig. 16A) lateral view. Nevertheless, clavidiscs are synapomorphic for *Merlia* and believed to be sigmancistra derivatives (*Hooper & Van Soest, 2002*), contrary to the spherancora's chelae origin. Coincidentally, the lateral view of cleistochelae (Fig. 16B) has also been misinterpreted as spherancorae, with which they share their chelae origin and the presence of partially fused alae. However, cleistochelae lack the inner teeth-brims and present a single arc (2D byplan), resulting from the fusion of all free alae in a single piece, whereas spherancorae present two arcs (3D byplan), as they result from the fusion of each one of the free alae with its opposing counterpart.

Finally, and despite their unique morphology amongst sponge microscleres, the function of spherancorae, as that of many other microscleres, remains unclear. In this sense, while megascleres possess a clear architectural role in the sponge skeleton, microscleres are mostly believed to play a consolidating or defensive role, if any

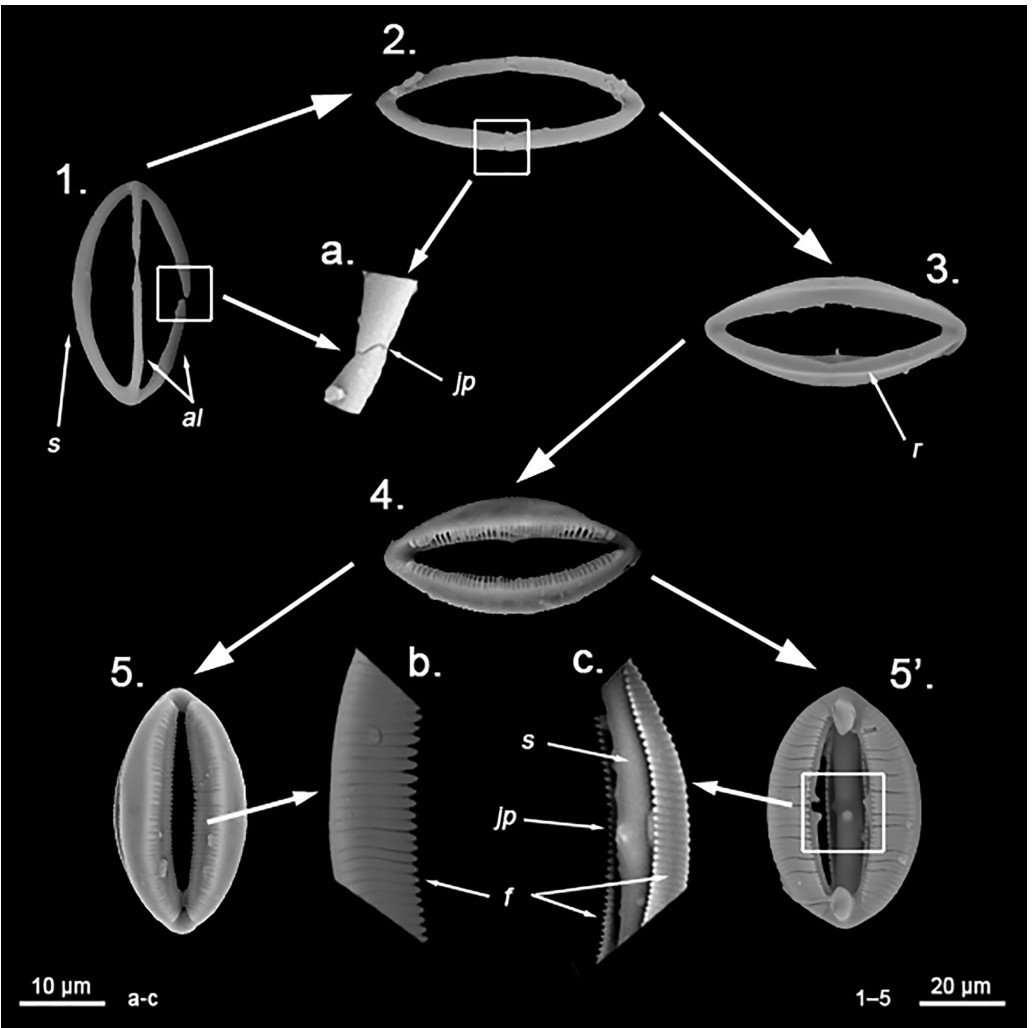

**Figure 15 Developmental stages of spherancorae.** Formation process of a spherancorae. 1. Initial stages of formation; the chelae origin can still be observed, with a full formed shaft (*s*) and free alae (*al*) still visible; 2. Fusion phase; the alae coalesce forming the four shafts; alae's junction points (*jp*) are visible (a.); 3. Thickening phase; the shafts start to thicken, and start forming the ridges (r) from which the fimbriae will later develop; 4. Fimbriae development phase; fimbriae start developing on the ridges, while the shafts continue thickening; 5. Fully formed spherancorae, with complete, free fimbriae (*f*) clearly visible (b.); 5'. Internal view of a spherancorae, visible due to the braking of a shaft; the junction point (*jp*) of the alae is still visible on the internal side of the shafts as a swelling (c.), while it is observable that fimbriae (*f*) are mostly free, only attached to the shafts (*s*) by its base. Scale bar for Figures 1–5 is 20 µm, whereas for figures a., b., and c. is 10 µm. All images were taken from *Melonanchora tumultuosa* sp. nov. (NHMUK Norman Coll. 1898.5.7.38).

(*Uriz et al., 2003*). In the *M. elliptica* holotype, spherancorae were observed to concentrate and form a dense palisade on the outer layer of the choanosome as well as surrounding the aquiferous canals, which could imply a defensive role, or a possible role in the architecture of the aquiferous system, yet this was not observed in any other of the samples analysed, and remains speculative.

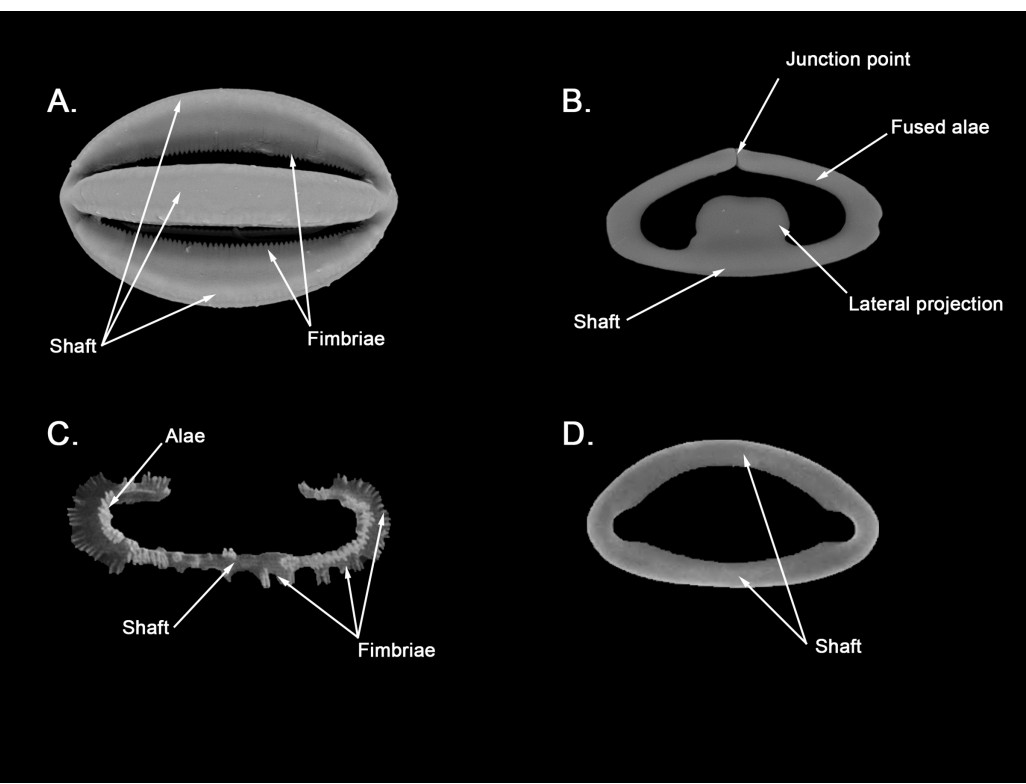

**Figure 16 Comparison between spherancorae and other fused chelae.** (A) Spherancorae from *Melonanchora elliptica* (NHMUK 1882.7.28.54a); (B) cleistochelae from *Clathria* sp. (NHMUK 1910.10.12.18); (C) placochelae from *Guitarra dendyi* (Kirkpatrick, 1907) (Ríos pers. Coll.); (D) Clavidisc from *Merlia normani* Kirkpatrick, 1908 (Uriz pers. Coll.).

## ACKNOWLEDGEMENTS

The authors would like to thank Inés Fernández, Alejandra Calvo and Cristina Boza from the IEO Gijón for their help with samples from the Galician Bank, as well as Mar Sacau, from the IEO Vigo for allowing us to use the material from the NEREIDA surveys, Inês Gregório (CIIMAR) for her support with the molecular work performed and lastly, Christine Morrow for critically reviewing the manuscript's English use. Additionally, the first author would like to thank Maria Pascual for her invaluable hospitality in Gijón; Tetiana Stefanska for her help getting access to Russian literature; Patricia Baena and Marina Biel for their help with some samples; Carlota Ruiz, for listening to endless conjecturations and her aid with bibliography; Jordi Grinyó, for always being there with a helping hand; Alfredo Quintana (University of Oviedo), María García (CEAB-CSIC) and José Manuel Fortuño (ICM-CSIC) for their technical assistance during SEM image acquisition and finally, to all the personnel in the Centro Oceangráfico de Gijón (IEO) for their warmth during his stay there. The authors would also like to thank the inestimable help of all the museum's curators and staff: Dr. Tom White from the NHMUK, Maria Taviano, from the MSNG, Dr. Jean-Marc Gagnon from the CMNI, Dr. Maria Mostadius from the MZLU, Dr. Marie Meister from the MZS, Dr. Carsten Lüter from the ZMB, Allen Collins and Lisa Comer from the NMNH, Eric A. Lazo-Wasem from the YPM, Bram

van der Bijl from the NBC, Manuel R. Solórzano for providing samples from Galicia, and last, but not least, Prof. Jean Vacelet, whom allowed us access to his personal collections, for which the authors are very grateful of.

### Funding

This research has been performed in the scope of the SponGES project, which received funding from the European Union's Horizon 2020 Research and Innovation Programme under grant agreement no. 679849. This study was funded by the European Commission LIFE C "Nature and Biodiversity" call, and included in the INDEMARES (07/NAT/E/000732) and INTEMARES (LIFE15 IPE ES 012) projects. The Biodiversity Foundation, of the Ministry of Environment, was the institution responsible for coordination these projects. The present investigation was undertaken as part of the NAFO Potential Vulnerable Marine Ecosystems-Impacts of Deep-Sea Fisheries project (NEREIDA) (Grant Agreement S12.770786), which is supported by Spain's General Secretary of the Sea (SGM), Spain's Ministry for the Rural and Marine Environment, the Spanish Institute of Oceanography, the Geological Survey of Canada, the Canadian Hydrographic Service, Fisheries and Oceans Canada, the UK's Centre for the Environment Fisheries and Aquaculture Science (Cefas), the Russian Polar Research Institute of Marine Fisheries and Oceanography, and the Russian P.P. Shirshov Institute of Oceanology (RAS). Sample MZB 2019–1740 was collected under the ABRIC project (Ref. RTI2018-096434-B-I00) funded by the Spanish Ministry of Science and Innovation. Finally, AS was the recipient of the 2019 Young Scientist Best Paper Award of the Dept. of Marine Biology and Oceanography at the Institute of Marine Sciences, which provided funding for the SEM imaging, alongside with the grant Consolidate SGR378 Benthic Ecology from the Generalitat de Catalunya awarded to MJU with the institutional support of the 'Severo Ochoa Centre of Excellence' accreditation (CEX2019-000928-S). There was no additional external funding received for this study. The funders had no role in study design, data collection and analysis, decision to publish, or preparation of the manuscript.

### Grant Disclosures

The following grant information was disclosed by the authors:
European Union's Horizon 2020 Research and Innovation Programme: 679849.
European Commission LIFE C "Nature and Biodiversity" call.
INDEMARES: 07/NAT/E/000732.
INTEMARES: LIFE15 IPE ES 012.
NAFO Potential Vulnerable Marine Ecosystems-Impacts of Deep-Sea Fisheries project (NEREIDA): S12.770786.
Spain's General Secretary of the Sea (SGM).
Spain's Ministry for the Rural and Marine Environment.
Spanish Institute of Oceanography.
Geological Survey of Canada.

Canadian Hydrographic Service.
Fisheries and Oceans Canada.
UK's Centre for the Environment Fisheries and Aquaculture Science (Cefas).
Russian Polar Research Institute of Marine Fisheries and Oceanography.
Russian P. P. Shirshov Institute of Oceanology (RAS).
ABRIC project: Ref. RTI2018-096434-B-I00.
Spanish Ministry of Science and Innovation.
Institute of Marine Sciences.
Generalitat de Catalunya: SGR378.
Severo Ochoa Centre of Excellence: CEX2019-000928-S.

## Competing Interests

The authors declare that they have no competing interests.

## Author Contributions

- Andreu Santín conceived and designed the experiments, performed the experiments, analyzed the data, prepared figures and/or tables, authored or reviewed drafts of the paper, and approved the final draft.
- María-Jesús Uriz conceived and designed the experiments, performed the experiments, analyzed the data, authored or reviewed drafts of the paper, and approved the final draft.
- Javier Cristobo performed the experiments, analyzed the data, authored or reviewed drafts of the paper, and approved the final draft.
- Joana R Xavier performed the experiments, analyzed the data, authored or reviewed drafts of the paper, and approved the final draft.
- Pilar Ríos performed the experiments, analyzed the data, authored or reviewed drafts of the paper, and approved the final draft.

## Data Availability

The specimen are reposited in the following locations:

National History Museum: NHMUK 1882.7.28.54a; NHMUK - Norman Coll. N°50 10.1.1.1417; NHMUK 1954.3.9.301 N°50; NHMUK-Norman Coll. -H. J. Carter Slide Coll. 1954.3.9.301; NHMUK -Norman Collection 1910.1.1.588; NHMUK-Sott-Ryen Coll., 1931.6.1.19; NHMUK Norman Coll. 1910.1.1.1418; NHMUK–Norman Coll. 1910.1.1.1419; NHMUK–Norman Coll. 1910.1.1.1420; NHMUK–Norman Coll. 1910.1.1.1421; NHMUK–Norwegian Coll. 1982.9.6.14.a.; NHMUK–Icelandic Coll. 1958.1.1.633; NHMUK Norman Coll. 1898.5.7.38; NHMUK, 83.12.13.70.89; NHMUK 1963.7.29.23

Museum für Naturkunde: ZMB Por 3042; ZMB Por 2680; ZMB Por 6571.

Canadian Museum of Nature: CMNI 2018-0107.

Museum of Biology of Lund: MZLU L936/3483; MZLU L935/3858.

Swedish Museum of Natural History: NRM 113070.

Yale Peabody Museum of Natural History: YPM IZ 006552.PR.

Naturalis Biodiversity Center: ZMA.POR.P.10797; ZMA.POR.1548; ZMA.POR.
P.10800; ZMA.POR.20192; ZMA.POR.P.10799; ZMA.POR.20559.b; ZMA.POR.20473.b;
ZMA.POR.20551; ZMA.POR.P.10798; ZMA.POR.20353.a; ZMA.POR.P.10795; ZMA.
POR.P.20020; ZMA.POR.20020; ZMA.POR.P.10829; ZMA.POR.20467; ZMA.POR.
P.10828; ZMA.POR.20175.b; ZMA.POR.P.10827; ZMA.POR.20335; ZMA.POR.P.10796;
ZMA.POR.P.10825; ZMA.POR.P.10822; ZMA.POR.P.10824; ZMA.POR.4977; ZMA.POR.
P.10823; ZMA.POR.4976; ZMA.POR.7269; ZMA.POR.P. 10826

Gothenburg Natural History Museum: GNM Porifera 416; GNM Porifera 290; GNM
Porifera 390; GNM Porifera 624.

Museo Civico di Storia Naturale di Genova: MSNG Vis4.7.

Museu de Ciències Naturals (Zoologia) de Barcelona: MZB 2019–1740.

Musée Zoologique de la Ville de Strasbourg: MZS Po165.

National Museum of Natural History, Smithsonian Institution: NMNH-USNM
1082996; NMNH-USNM 148959; NMNH-USNM 1478958.

Additional information on each of the analyzed specimens can be found on the species
descriptions.

## New Species Registration

The following information was supplied regarding the registration of a newly described
species:

Publication LSID: lsid:zoobank.org:pub:F1A22CAA-DE1F-434D-9A6B-F00853C40FF5

*Arhythmata* gen. nov.: urn:lsid:zoobank.org:act:25BD6F3B-D818-432B-854C-
71AB54DB72BA

*Hanstoreia* gen. nov.:
urn:lsid:zoobank.org:act:B89C3FC5-8EDA-4D53-9FFC-9417E75E97E7

*Melonanchora insulsa* sp. nov.: urn:lsid:zoobank.org:act:1082F7BF-4584-47A3-B3E5-
E0E8E81CC9E7

*Melonanchora intermedia* sp. nov.: urn:lsid:zoobank.org:act:986651DC-2C16-4EC8-
8D66-D28FC963EBB0

*Melonanchora maeli* sp. nov.: urn:lsid:zoobank.org:act:48A1B8FE-9C46-4D9C-986E-
A8EDF6383537

*Melonanchora tumultuosa* sp. nov.: urn:lsid:zoobank.org:act:1E7784B2-9BBE-4854-
9BDF-0616D8BA8F0A

## Supplemental Information

Supplemental information for this article can be found online at http://dx.doi.org/10.7717/
peerj.12515#supplemental-information.

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
