# Peer review of "Unique spicules may confound species differentiation: taxonomy and biogeography of Melonanchora Carter, 1874 and two new related genera (Myxillidae: Poecilosclerida) from the Okhotsk Sea"

_PeerJ, doi:10.7717/peerj.12515_

## Round 0.1 · original submission · Minor Revisions

Dear Dr Santín,

We have received the reports from our reviewers on your manuscript.

Based on the advice received, I feel that your manuscript could be accepted for publication should you be prepared to incorporate minor revisions.

When preparing your revised manuscript, you are asked to carefully consider the reviewers' comments which are attached, and submit a list of responses to the comments. Both reviewers have noticed that your manuscript is not good enough in English, so pay special attention to this remark and find a native English speaker. Your list of responses should be uploaded as a file in addition to your revised manuscript.

Best regards,
Alexander Ereskovsky

·

Basic reporting

The MS is a great piece of work, well written and clearly argued, but could benefit from improving the English expression, numerous misspellings and incorrect word usage. I suggest that the authors engage a professional English language expert to check the MS before re-submission.

The MS is referenced well and sufficient background and context is supplied

The raw data appears to be in Table 1 and I have comments on the difficult of the format below
The Figures are attractive

Experimental design

The manuscript reviews the status of species assigned to poecilosclerid genus Melonanchora Carter, finding that the three species previously thought to be Okhotsk Sea endemics belong to different genera, one of them new, and new Mediterranean species are described on the basis of differences in the megascleres as well as the microscleres.

I cannot fault the taxonomic process and many of the problems are minor and simply style issues that relate to the way the primary author writes. They are listed in detail in the Additional comments box below

The research questions are important for conservation and the results fill an identified gap in a sponge biodiversity knowledge

Validity of the findings

The majority of findings appear to be valid, however I struggled with assessing the veracity of the taxonomic decisions in the “Remarks” sections of each new species description (e.g., that the species was new, and differs from this and that species, etc) because most of the decisions require the reader to struggle through Table 1 to find the individual spicule dimensions for the compared species (note that the “this study” specimen details are not included which makes it even harder), and the various figures, when comparing the current species with others. It is standard practise that all such comparisons are presented clearly within the “Remarks” sections, stating and comparing in one sentence the key dimensions for the compared species, all in one place, saving a huge amount of time for the reader. It is not adequate to just say, “ this and that differed in shape and dimensions”. The inclusion of all this data, in a well-argued paragraph, makes it easier for the reader to follow the co-authors decision to assign a new species; at the moment the reader has to assess the integrity of such new species by gathering all the data together to make the own, personal decision.

Additional comments

Line 146 – 155 I would advise adding the actual museum collection accession prefix for specimens listed in Section 2.1 Museum material and sample treatment because they often differ from the abbreviations you currently have. For example: Museum für Naturkunde, previously known as Zoologisches Museum Berlin, Germany (ZMB) yet the specimens are listed as ZMB Por 3042 (Line 267). Here is a suggestion as to a clear style of writing might look like (from Reiswig & Kelly, 2018): Registration of type and general material. Primary and secondary type materials of new species, and additional material, are deposited in the NIWA Invertebrate Collection (NIC) at the National Institute of Water and Atmospheric Research (NIWA; formerly New Zealand Oceanographic Institute, NZOI), Greta Point, Wellington, using the prefix NIWA―, and the Museum of New Zealand Te Papa Tongarewa (formerly National Museum of New Zealand, NMNZ), using the prefix NMNZ PO.―.

Line 242 – You have established the new subgenus, Melonanchora (Toretendalia) subgen. nov., which necessitates the type species being set up in its own, new, subgenus Melonanchora (Melonanchora) Carter, 1874 subgen. nov. Please add “subgen. nov.” after the subgenus on line 242 so that it reads: Subgenus Melonanchora Carter, 1874 subgen. nov.

Line 242 – You say in the Remarks section to “see remarks for M(T) subgen. nov. It is preferable that the remarks that relate to Melonanchora (Melonanchora) Carter, 1874 subgen. nov. are given here, rather than making the reader try to find these former comments. Please bring them into the remarks section here.

Lines 268 & 270 (and throughout) – It is usual to write the latitudes and longitudes in the decimal format currently. Unless you can provide a good reason for not doing this (e.g. the record is a historical record that you have not collected), then it would be good to see these done throughout.

Line 265 – It is usual to italicise the name of collecting vessels. Please write the name of the vessel (and all others) as HMS Porcupine, without the quotes. Please check throughout MS.

Line 305 – ‘..any clearly discernible patter.’ Pattern

Line 338 – M. elliptica is a common…..Never abbreviate a genus name that starts a sentence

Line 354 – ‘….collected during the ‘Porcupine’ expedition in…’ Please check the details and change this to ‘…collected during the HMS Porcupine Expedition (1869) in the Northwest Atlantic (Carter….’. Please also check this annotation throughout the MS.

Line 595 –599 ‘…Additionally, it has been suggested that…’ Is this really the best place for this statement? Wouldn’t it be better up with M. cf emphysema, drawing in the environmental argument into that species? The statement here is distracting even though I can see why you are adding it in (to strengthen your argument that tumultosa is not elliptica even though they co-occur), but it is not relevant to tumultosa at this stage?

Line 606 – If M(M) intermedia is closest to M(M) emphysema – shouldn’t you place this species description just after the latter for a better comparison?

Line 659 – I note that most of the species “Remarks” sections for the species descriptions require the reader to struggle through Table 1 to find the comparative spicule dimensions for species (note that the “this study” specimen details are not included which makes it even harder), and the various figures, when comparing the current species with others.

It is standard practise that all such comparisons are presented clearly within the “Remarks” sections, outlining, and stating the key dimensions for the compared species, all in one place, saving a huge amount of time for the reader.

A good example is Line 724-725: “Differences in shape and size between microscleres of both species support that M. insulsa is a different species from M. elliptica.”

It would be much better to read about the dimension differences right here so that I don’t have to go to Table 1 and find the dimensions. This also makes it easier for the reader to assess the integrity of such new species -the reader is left to get all the data together for themselves to make the decision. Please consider expanding these sections.

Lines 831-847 Integrity of Melonanchora (Toretendalia) globogilva Lehnert, Stone & Heimler, 2006a, from the Aleutian Islands. I can see you have struggled a little with the decision to erect a new subgenus or a new genus for this unusual species. Although this new species shares several morphological characters with M(M), it is the only Pacific representative of the genus (thus extremely rare) and it has full acanthostyles (that don’t appear to be simply modified styles) and VERY unusual box-like spicules that, I think, are no more close to spherancoras than placochelae of Placospongia and anisoplacochelae of Asbestopluma anisoplacochela! However, that is for you to judge. With a slightly more indepth comparison with these other genera, it might be possible to propose a new genus, Toretendalia gen. nov., as there is (in my opinion) enough to work on. This would require a little more information on the skeletal differentiation between the genera. Something to consider anyway. I will leave this decision to you.

Line 1084 – “Skeleton: Typical of the genus”. This looks like a short-cut! – please explain the skeleton architecture as you have for everything else and this is the type taxon.

Line 1105 – These look remarkably like Abyssochelae! No response required but might be worth a comment

Table 1 – While this is indeed an impressive table, it would be very helpful to have the specimen accession number in column 1 as there is no clear reference back to your study – and very frustrating if you are trying to track a specific specimen measurement.

·

Basic reporting

Language requires some more work in terms of both spelling and grammar, marked up directly on the MS PDF, but would benefit from a final check by a native English speaker once referees comments/ corrections are attended to, and before the MS is resubmitted for publication.
The relevant literature is thoroughly reviewed and exhaustedly documented, most of which is required to support the authors’ decisions. Ideally the literature citations should include DOIs (or BHL) digital links as required my many journals nowadays, where available, but I am not sure if this is a requirement of PeerJ.
The illustrations are of reasonable quality and all are necessary to support the taxonomic descriptions and decisions made by the authors.
The Table is rather long and might be difficult to publish, and might be more easily reported as a series of smaller tables?

Experimental design

The methodology is standard for these sorts of taxonomic papers, and adequately reported in the MS. One outstanding feature of this MS is the depth and breadth of the material examined from multiple collections, making it a top class, comprehensive taxonomic revision and highly important to this particular group of myxillids as a benchmark for future research on these sponges.
As mentioned elsewhere, despite the authors’ attempts, they were unable to get quality DNA to sequence the “standard barcoding” CO1 gene, which is a weakness but not uncommon with old collections with poor preservation history, and apparently for sponges in general.

Validity of the findings

The taxonomic conclusions are well-supported by both raw and analysed data/ specimens. The biogeographical component is somewhat speculative, based mostly on the literature, but without an unequivocal fossil record and molecular data, this probably can’t be improved – nevertheless, I recommend leaving it in the MS as it is of interest and as an hypothesis yet to be thoroughly tested in the future.

Additional comments

This is an outstanding, thoroughly researched contribution to the taxonomy of a relatively small group of geographically restricted myxillid sponges. It is based on accumulating and thoroughly redescribing a large collection from multiple museums and private collections, leading to a significant revision of known species, new species and new genera and subgenera. In my opinion the taxonomic decisions made by the authors are justifiable and well-defendable, and unfortunately despite their attempts to sequence old collections, would have been more strongly supported with molecular sequence data (a common problem in sponge taxonomy based on old material). Hopefully, at some time in the future, our technologies will have evolved to overcome this problem.

---

## Round 0.2 · accepted · Accept

Thank you for your thorough work on correcting the first version of the article and detailed answers to the reviewers. The new version of the manuscript satisfies all the comments of the reviewers.